# LOCALITY SENSITIVE AVATARS FROM VIDEO

**Chunjin Song**[1]*, **Zhijie Wu**[1]*, **Shih-Yang Su**[1], **Bastian Wandt**[3], **Leonid Sigal**[1,2] **& Helge Rhodin**[1,4]
[1]Department of Computer Science, University of British Columbia      [2]Vector Institute for AI
[3]Department of Electrical Engineer, Linköping University     [4]Bielefeld University
`{chunjins,lsigal,rhodin}@cs.ubc.ca`

## ABSTRACT

We present locality-sensitive avatar, a neural radiance field (NeRF) based network to learn human motions from monocular videos. To this end, we estimate a canonical representation between different frames of a video with a non-linear mapping from observation to canonical space, which we decompose into a skeletal rigid motion and a non-rigid counterpart. Our key contribution is to retain fine-grained details by modeling the non-rigid part with a graph neural network (GNN) that keeps the pose information local to neighboring body parts. Compared to former canonical representation based methods which solely operate on the coordinate space of a whole shape, our locality-sensitive motion modeling can reproduce both realistic shape contours and vivid fine-grained details. We evaluate on ZJU-MoCap, SynWild, ActorsHQ, MVHumanNet and various outdoor videos. The experiments reveal that with the locality sensitive deformation to canonical feature space, we are the first to achieve state-of-the-art results across novel view synthesis, novel pose animation and 3D shape reconstruction simultaneously. Our code is available at `https://github.com/ChunjinSong/lsavatar`.

## 1 INTRODUCTION

Reconstructing a photorealistic human avatar is crucial for various downstream applications (Wu et al., 2019; Bagautdinov et al., 2021; Peng et al., 2021a; Lombardi et al., 2021; Hu et al., 2022; Song et al., 2023). Some previous methods achieve this by rigging a surface mesh to an underlying skeleton and enhancing the mesh with either a neural texture (Bagautdinov et al., 2021; Liu et al., 2021) or learnable vertex features (Kwon et al., 2021; Peng et al., 2021a;b). While these approaches offer detailed reconstructions, they are prone to producing artifacts when learning from sparse examples. Other efforts (Bagautdinov et al., 2021; Xu et al., 2021; Wang et al., 2022a) use a parametric template to mitigate this issue but are limited in representing various human shapes and poses. They are also difficult to optimize and often yield artifacts due to their explicit handling of vertices and faces.

Recently, significant attention has been devoted to extending Neural Radiance Field (NeRF) models (Su et al., 2021; 2022; Weng et al., 2022) for dynamic avatar modeling using a human skeleton as input. Specifically, A-NeRF (Su et al., 2021) and NARF (Noguchi et al., 2021) compute relative coordinates of the input query points in relation to skeletal joints and then estimate geometry and appearance information by using these coordinates holistically. Subsequently, DANBO (Su et al., 2022), Posevocab (Li et al., 2023) and NPC (Su et al., 2023) decompose the feature space into parts and aggregate these features before computing the final outputs, improving network scalability. PM-Avatar (Song et al., 2024a) has further advanced this by modulating the frequency transformation of query points through modeling part relationships, enhancing fine-grained detail synthesis. Recently, the 3D Gaussian Splatting (3DGS) (Kerbl et al., 2023a) based methods leverage a set of anisotropic Gaussian functions to encode local patterns and transform the template Gaussians to match the per-frame input images for fast rendering (Qian et al., 2024; Wen et al., 2024; Lei et al., 2024).

Despite the aforementioned empirical advances, these methods struggle to generalize well and yield artifacts with sparse input data, such as an in-the-wild video from the online Youtube platform. A key issue of these methods is that they render the final outputs in the observed pose spaces defined

---

* Equal Contribution.

| | Monocular input | Locality encoding | Local deformation | Canonical space rendering | Geometry Reconstruction |
|---|---|---|---|---|---|
| HumanNeRF (CVPR 2022) | ✓ | ✗ | ✗ | ✓ | ✗ |
| MonoHuman (CVPR 2023) | ✓ | ✗ | ✗ | ✓ | ✗ |
| Vid2Avatar (CVPR 2023) | ✓ | ✗ | ✗ | ✓ | ✓ |
| NPC (ICCV 2023) | ✗ | ✓ | ✗ | ✗ | ✗ |
| PM-Avatar (ICLR 2024) | ✗ | ✓ | ✗ | ✗ | ✗ |
| 3DGS-Avatar (CVPR 2024) | ✓ | ✓ | ✗ | ✗ | ✗ |
| GoMAvatar (CVPR 2024) | ✓ | ✓ | ✗ | ✗ | ✓ |
| GauHuman (CVPR 2024) | ✓ | ✓ | ✗ | ✗ | ✗ |
| GaussianAvatar (CVPR 2024) | ✓ | ✓ | ✗ | ✗ | ✗ |
| **Ours** | ✓ | ✓ | ✓ | ✓ | ✓ |

Table 1: **Comparison to SOTA.** Our full model conceptually unifies the idea of rendering in the canonical pose space with local pattern encoding for deformations, improving generalization and adaptive detail synthesis in monocular settings. Signed Distance Function (SDF) is also learned for explicit geometry reconstruction. We refer Sec. C and Fig. B for more discussions.

by different inputs. Thus the rendered images cannot be directly derived from the pose-independent information for generalization to testing camera views and unseen poses.

Here we aim to reconstruct human representations with intricate and dynamic details from monocular video sequences. To this end, state-of-the-art methods (Jiang et al., 2022; Li et al., 2022; Wang et al., 2022a) build a body model in the canonical pose space rather than in observed pose spaces, such that the geometry and appearance can be learned statically. Specifically, (Guo et al., 2023) directly transforms the input skeleton and the positions in canonical pose space to shapes and colors for rendering. Meanwhile, (Weng et al., 2022; Yu et al., 2023) explicitly model a skeletal rigid motion field together with its corresponding non-rigid motion field to faithfully capture the dynamically changing details (e.g. cloth wrinkles). While these methods successfully estimate a stable avatar representation, their learning pipeline is purely conditioned on the entire skeleton pose. Thus they are prone to poor generalization when unrelated body parts remain weakly entangled as shown in Fig. B.

In this paper, we introduce *Locality Sensitive Avatar*, which is a novel deformation based approach to directly disentangle independent body parts and learn a generalizable personalized human model from monocular videos. Tab. 1 summarizes connections between our approach and existing algorithms. In detail, we decompose movements into a skeletal rigid motion to the canonical pose space and a corresponding non-rigid counterpart, which are learned locally. Specifically, we apply inverse linear blending skinning (LBS) for rigid motion, followed by generating point-dependent offsets in canonical space, with features encoding local pose context as the non-rigid component. Part relationships are approximated via a graph neural network (GNN) to account for the irregular structure of the human skeleton. This approach enables multi-scale deformations by linking rigid motion to holistic modeling and localizing non-rigid deformations, allowing us to simultaneously learn invariant avatar representations and high-frequency details. In contrast, previous methods handle deformations monolithically or implicitly, resulting in suboptimal performance. We also apply the Signed Distance Function (SDF) to enhance detailed geometry reconstruction. Hereafter, we refer to canonical pose space as canonical space and observed pose space as observation space for brevity.

We evaluate our method on three types of datasets: synthetic examples, existing lab-made sequences, and videos captured outdoors. Our contributions can be summarized as following:

- We introduce a novel locality-sensitive avatar representation by integrating part-based locality encoding with canonical space rendering.
- We utilize graph neural network (GNN) features to capture local part contexts and then estimate non-rigid deformation offsets, enabling the generation of adaptive details.
- We demonstrate significant improvements over state-of-the-art methods in novel view synthesis, unseen pose rendering and shape reconstruction simultaneously.

## 2 RELATED WORK

Novel image rendering for human avatars recreates complex geometry and subtle lighting effects appeared in given image sequences. Recently, a lot of efforts have been devoted to apply neural

radiance field for image-based rendering (Xie et al., 2022). Here we only review the most related methods in radiance field based rendering and neural avatar modeling.

**Neural Rendering.** In computer graphics and 3D vision (Park et al., 2019; Wu et al., 2019; Mescheder et al., 2019; Takikawa et al., 2021; Müller et al., 2022; Zhang et al., 2024), neural fields are extensively studied due to their impressive performance and give birth to Neural Radiance Fields (NeRF) (Mildenhall et al., 2020), a strong paradigm to learn 3D representations from 2D images for novel view synthesis. Since then, numerous efforts have been made to adapt NeRF to dynamic scenes by additionally incorporating the time dimension (Li et al., 2021b; Park et al., 2021; Pumarola et al., 2021). On the other hand, some works replace the density outputs of NeRF models with Signed Distance Function (SDF) values to enable detailed geometry modeling (Wang et al., 2021; Yariv et al., 2021; Wang et al., 2022b). To alleviate the time-consuming volumetric rendering operations used in NeRF and its follow-ups approaches, the recent 3D Gaussian Splatting (3D-GS) framework (Kerbl et al., 2023a) applies explicit Gaussian primitives for scene modeling and projects the learned 3D Gaussians to the 2D image plane for fast visual synthesis. Later on, (Luiten et al., 2024; Wu et al., 2024; Kerbl et al., 2023b) extend the 3D-GS pipeline to dynamic scenes by learning time-dependent Gaussian parameters. Different from these methods, we focus on modeling an animatable avatar via NeRF framework by restricting the non-rigid motion to be localized for adaptive detail generation.

**Modeling Articulated Avatars.** Pioneer works in human body modelling apply SMPL (Loper et al., 2015; Pavlakos et al., 2019) as the mesh template to provide a parametric representation of the human body shape. Recently, literature explores the use of neural representations, either based on NeRF or SDF to circumvent the SMPL prior. Some works (Su et al., 2021; Noguchi et al., 2021; Su et al., 2022; 2023; Song et al., 2024a) compute the relative coordinates of input query points and estimate the density and color values through a NeRF model as they hypothesize that each part approximately remains static during avatar deformation. Meanwhile, other works (Peng et al., 2021a; Wang et al., 2022a; Li et al., 2022; 2023) define a static radiance field in the coordinate space of canonical pose and warp the ray marching frame by frame. To reconstruct detailed 3D avatars from monocular videos, HumanNeRF (Weng et al., 2022) and MonoHuman (Yu et al., 2023) explicitly decompose the motion field into skeletal rigid and corresponding non-rigid motions, as in previous disentanglement based methods (Huang et al., 2018; Lee et al., 2018; Song et al., 2019; Wu et al., 2020; 2022; Wang et al., 2024). Concurrently, (Guo et al., 2023; Song et al., 2024b) directly map the input poses and positions in canonical space to SDF and color values for volume rendering. Recent efforts (Zielonka et al., 2023; Lei et al., 2024; Moreau et al., 2024; Kocabas et al., 2024a; Qian et al., 2024; Wen et al., 2024; Hu et al., 2024a;b; Kocabas et al., 2024b) combine 3D-GS framework with human articulation prior for fast avatar reconstruction. Given these progress, none of the aforementioned deformation-based approaches consider the locality modeling and deformations to the canonical pose simultaneously. Thus existing baselines either focus on reconstructing a sequence at high detail or generalizing well under novel settings; but not both. See Tab. 1 for better conceptual comparisons.

## 3 METHOD

Fig. 1 illustrates our network which learns to reconstruct a 3D articulated avatar from a monocular video. Our key contribution is a locality sensitive deformation network to compute an offset in canonical space. Taken a human skeleton and a query point in per-frame observation space as input, we first apply a Graph Neural Network (GNN) to estimate local relationships between different body parts and compute a set of relative coordinates in each part's local frame. These relative coordinates will be combined with corresponding GNN features and then be aggregated into a locality-wise offset with a learnable window function. We mix the canonical position from skeletal deformation with its point-dependent offset and regress the SDF and color values for volume rendering.

### 3.1 SKELETAL DEFORMATION TO CANONICAL COORDINATE SPACE

It is crucial to learn an expressive canonical space for high-fidelity 3D avatar modeling from monocular videos. Existing methods (Weng et al., 2022; Yu et al., 2023; Guo et al., 2023) typically adopt the skeletal deformation, which is driven by input human skeleton, to transform the query point $x \in \mathbb{R}^3$ in observation space to $x_c$ in canonical space. Here we define the pose with $N$ parts as $\theta = [\mathbf{B_1}, \mathbf{B_2}, \ldots, \mathbf{B_N}]$ and $\mathbf{B_i} \in \mathbb{R}^6$ (Zhou et al., 2019) represents the rotation parameter of $i$-th

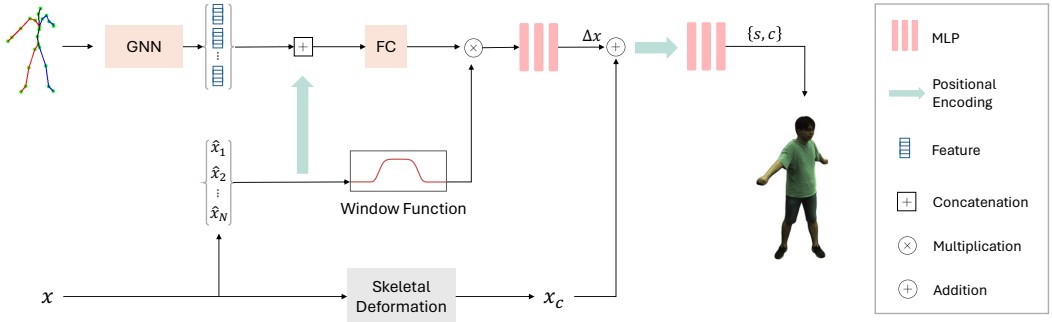

Figure 1: **Architecture overview**. Given the query point $x$, we first perform skeletal deformation to compute its canonical coordinate $x_c$. Next, $x$ is transformed into a set of relative positions $\{\hat{x}_i\}$ within each part's local frame. With a learnable window function, we then aggregate the concatenations of these per-part relative positions with their corresponding part features from the Graph Neural Network (GNN) into a single feature to estimate a coordinate offset $\triangle x$ as the non-rigid motion. Finally, we input the sum of $x_c$ and $\triangle x$ into a Multilayer Perceptron (MLP) to regress the Signed Distance Function (SDF) value $s$ and color $c$ for SDF-based volume rendering.

part. Following Vid2Avatar (Guo et al., 2023), we perform the inverse of linear-blend skinning to connect $x_c$ in canonical space to $x$ in deformed space as

$$x_c = \left( \sum_{i=1}^{N} \tilde{\omega}_i \mathbf{B_i} \right)^{-1} x. \tag{1}$$

Here $\{\tilde{\omega}_i\}$ denotes the skinning weight of the $i$-th bone and is associated with the average of the nearest SMPL vertices' skinning weights. Note that, HumanNeRF (Weng et al., 2022) and MonoHuman (Yu et al., 2023) turn to use a set of learnable weight volumes with faster coordinate transformation to perform the inverse of linear-blend skinning. However, we find that the SMLP-based weight computation can improve the GPU memory consumption and convergence speed experimentally.

## 3.2 LOCALITY SENSITIVE DEFORMATION FIELD

As depicted in Fig. 1, we propose a novel deformation network to compute part adaptive offsets as the non-rigid motion representation. Sharing the same spirits with DANBO (Su et al., 2022), NPC (Su et al., 2023) and PM-Avatar (Song et al., 2024a), we regard the human skeleton as a graph, where each joint is linked to a graph node. Thus we apply the graph neural network (GNN) to model the relationships between skeleton parts as:

$$[G_1, G_2, \ldots, G_N] = \text{GNN}(\theta), \tag{2}$$

where GNN and $G_i$ represent a learnable graph neural network and the feature vector of the $i$-th part respectively. We adopt individual multi-layer perceptron (MLP) weights for each graph node such that the irregular nature of human skeleton can be better considered.

To learn the locality sensitive deformation, we transform a point $x$ to the local frame of $i$-th part as

$$\begin{bmatrix} \hat{x}_i \\ 1 \end{bmatrix} = T(\mathbf{B_i}) \begin{bmatrix} x \\ 1 \end{bmatrix}. \tag{3}$$

$T(\mathbf{B_i})$ here means the world-to-bone coordinates transformation matrix defined by the rotation parameter $\mathbf{B_i}$. Then we feed the local coordinate set $\{\hat{x}_1, \hat{x}_2, \cdots, \hat{x}_N\}$ into a positional encoding $\tilde{\gamma}(\cdot)$ (Mildenhall et al., 2020) respectively as

$$\tilde{\gamma}(\hat{x}_i) = [\hat{x}_i, \sin(2^0 \pi \cdot \hat{x}_i), \cos(2^0 \pi \cdot \hat{x}_i), \ldots, \sin(2^{L_{nr}-1} \pi \cdot \hat{x}_i), \cos(2^{L_{nr}-1} \pi \cdot \hat{x}_i)], \tag{4}$$

where $L_{nr}$ indicates the highest mapping frequency in $\tilde{\gamma}(\hat{x}_i)$. The concatenations of the frequency expanded features $\{\tilde{\gamma}(\hat{x}_i)\}$ and their part-wise matched GNN features $\{G_i\}$ are further processed by a per-bone individual fully connected layer as

$$f_i = \Phi_i([\tilde{\gamma}(\hat{x}_i), \ G_i]), \tag{5}$$

where $\Phi_i$ denotes the fully connected layer corresponding to $i$-th bone.

Similar to (Su et al., 2022; Song et al., 2024a), we leverage the window function to decide which per-part point-wise features to pass on to the downstream network and aggregate part features. To adaptively determine the valid range of each part, we compute scaled relative positions as $\{\bar{x}_i = s_i \cdot \hat{x}_i\}$ and $s_i$ is a learnable scaling coefficient of the $i$-th part which reflects the size of the volume that part makes effect. To jointly learn the volume dimensions $\{s_i\}$ and mitigate seam artifacts when parts overlap, we define the window function through $L_2$-norm $\|\cdot\|_2$ as

$$\bar{\omega}_i = \exp(-\alpha(\|\bar{x}_i\|_2^\beta)), \tag{6}$$

where we set $\alpha = 2$ and $\beta = 6$ across all experiments. Note that $\bar{\omega}_i$ can facilitate separating multiple volumes based on spatial affinities as it attenuates based on the relative spatial distances to the bone centers; see the function curve shown in Fig. 1. We normalize $\{\bar{\omega}_i\}$ to preserve the constant weight energy across different frames and yield the probability $p_i$, which control how much the $i$-th part contributes to non-rigid motion. Then we aggregate all point-dependent part features as

$$\bar{h} = \sum_{i=1}^{N} p_i \cdot f_i, \quad p_i = \frac{\bar{\omega}_i}{\sum_{j=1}^{N} \bar{\omega}_j}. \tag{7}$$

The aggregated feature $\bar{h}$ is inputted into a MLP to output a 3D offset $\triangle x$. In principle, the canonical position $x_c$ represents the coarse rigid deformation while the offset $\triangle x$ provides the more non-rigid effects. Thus, inspired by HumanNeRF (Weng et al., 2022) and MonoHuman (Yu et al., 2023), we handle complex human movement by mixing the rigid motion with its non-rigid counterpart as

$$T_c = x_c + \triangle x. \tag{8}$$

Finally, we feed the deformed position $T_c$ to another positional encoding $\bar{\gamma}(\cdot)$, followed by a MLP to estimate SDF output $s$ and color output $c$ as in standard NeRF framework:

$$\bar{\gamma}(T_c) = [T_c, \sin(2^0 \pi \cdot T_c), \cos(2^0 \pi \cdot T_c), \ldots, \sin(2^{L_c-1} \pi \cdot T_c), \cos(2^{L_c-1} \pi \cdot T_c)], \tag{9}$$

$$s, \quad c = \text{MLP}(\bar{\gamma}(T_c)). \tag{10}$$

Here $L_c$ and $\text{MLP}(\cdot)$ denote the highest mapping frequency in $\bar{\gamma}(\cdot)$ and a MLP network respectively. We set $L_{nr} = 5$ and $L_c = 5$ across all experiments and perform a validness test for $\{\bar{x}_i\}$ to cater for processing efficiency and local pattern concentration near surfaces.

**Relation to baselines.** In concept, our locality sensitive deformation field can be regarded as a unified framework of part related encodings (Su et al., 2022; 2023; Song et al., 2024a) and works rendering in canonical pose space (Weng et al., 2022; Yu et al., 2023; Guo et al., 2023) to bridge these two worlds. Similar to part related encodings, we use the GNN features as a building block to measure bone correlations and regress per-part feature. While DANBO (Su et al., 2022), NPC (Su et al., 2023) and PM-Avatar (Song et al., 2024a) focus on estimating part-level features and the transformation frequencies of query points, we associate the point-dependent GNN features with a Cartesian offset as a non-rigid deformation in canonical space. Then we can explicitly apply the invariant information during avatar deformation to yield generalizable rendering outputs. Compared to global canonical NeRF representations, our locality sensitive motion offset can better adapt to distinct body parts with high variability, thus leading to superior detail synthesis. In terms of empirical results, the pure part related encodings are prone to run into local minimum and severely distort desired patterns. On the other hand, the complete global canonical representations fail to accurately capture the spatially varying patterns, where our method can show its advantages in both Fig. B and Fig. 8.

## 3.3 SDF-BASED VOLUME RENDERING

Following VolSDF (Yariv et al., 2021), we first transform the SDF output $s$ to the density as

$$\sigma = \frac{1}{\beta} \cdot \Psi_\beta(-s). \tag{11}$$

Here $\beta$ and $\Psi_\beta$ represent a learnable transformation parameter and the Cumulative Distribution Function (CDF) of the Laplace distribution with zero mean and $\beta$ scale respectively.

Since the output density and color signals $\{\sigma, c\}$ are estimated in the canonical coordinate space, we synthesize the final rendering images via a vanilla NeRF model like other deformation-based baselines (Weng et al., 2022; Yu et al., 2023; Guo et al., 2023). Specifically, we sample $n$ points along a ray $r = (o, v)$ with camera center $o$ and ray direction $v$. Different from (Yariv et al., 2021; Guo et al., 2023), we only perform the uniform sampling to speed up the whole training process. The integral of the volume rendering equation can be approximately computed as

$$\hat{C}(r) = \sum_{i=1}^{n} \mathcal{T}_i \left(1 - \exp(-\sigma_i \delta_i)\right) \mathbf{c}_i,$$
(12)

$$\mathcal{T}_i = \exp(-\sum_{j=1}^{i-1} \sigma_j \delta_j),$$
(13)

where $\hat{C}$ is the rendered image and $\delta_i$ is the distance between adjacent samples along a given ray.

Following the existing neural radiance rendering pipelines for human avatars (Su et al., 2021; 2022; Wang et al., 2022a; Song et al., 2024a), we apply the $L_1$ based reconstruction loss $\|\cdot\|_1$ between $\hat{C}(r)$ and ground truth images $C_{gt}$ according to the whole ray set $\mathfrak{R}$ as

$$\mathcal{L}_{\mathrm{rec}} = \sum_{\mathbf{r} \in \mathfrak{R}} \left\| \hat{C}(r) - C_{gt}(r) \right\|_1.$$
(14)

Besides $\mathcal{L}_{\mathrm{rec}}$, we employ the LPIPS (Zhang et al., 2018) metric for network training to advance the detail rendering and improve the generalization to slight misalignments and shading variation between rendering outputs and ground truths. We denote the perceptual loss as $\mathcal{L}_{\mathrm{LPIPS}}$.

Additionally, we find that the network is prone to overfit appeared high-frequency patterns and produce large non-rigid deformation offset. Thus, to improve the network generalization to different human poses, we regularize $\Delta x$ with

$$\mathcal{L}_{\Delta \mathrm{x}} = \|\Delta x\|_2^2.$$
(15)

Finally, we encourage the output $s$ to more accurately approximate a Signed Distance Function by enforcing network outputs to satisfy the Eikonal constraints in canonical space as

$$\mathcal{L}_{\mathrm{eik}} = \mathbb{E}_{x_c}(\left\| \frac{\mathrm{d}s}{\mathrm{d}x_c} \right\| - 1)^2.$$
(16)

Thus, given a video sequence of a human, we aim to optimize the following combined loss function:

$$\mathcal{L} = \mathcal{L}_{\mathrm{rec}} + \lambda_{\mathrm{eik}} \mathcal{L}_{\mathrm{eik}} + \lambda_{\mathrm{LPIPS}} \mathcal{L}_{\mathrm{LPIPS}} + \lambda_{\Delta \mathrm{x}} \mathcal{L}_{\Delta \mathrm{x}}.$$
(17)

$\lambda_{\mathrm{eik}}$, $\lambda_{\mathrm{LPIPS}}$ and $\lambda_{\Delta \mathrm{x}}$ are weights of Eikonal loss, LPIPS loss and offset regularization respectively.

## 4 RESULTS

We compare with state-of-the-art methods focusing on modeling avatar representations from monocular video, including NeRF-based HumanNeRF (Weng et al., 2022), MonoHuman (Yu et al., 2023), NPC (Su et al., 2023), Vid2Avatar (Guo et al., 2023), PM-Avatar (Song et al., 2024a), and 3D Gaussian Splatting based 3DGS-Avatar (Qian et al., 2024) and GoMAvatar (Wen et al., 2024). We perform the empirical evaluations by assessing the results of novel view synthesis, novel pose generalization and 3D shape reconstruction. We also conduct ablation studies to measure the importance of locality sensitive offsets, skeletal deformation and spatial window function. The supplementary video provides further animation demonstration. Source code will be available upon publication.

### 4.1 EXPERIMENTAL SETTINGS

We follow the evaluation protocal established by MonoHuman and GoMAvatar to choose eight sequences from the ZJU-Mocap dataset (Peng et al., 2021b) with the same dataset splits. We further

Table 2: **Overall quantitative comparisons.** While baseline methods excel individually in novel view synthesis, pose animation, and geometry reconstruction, our locality-sensitive avatar representation outperforms them across all three tasks, achieving over **10%** improvement in LPIPS, **20%** in FID, and **50%** in KID. Despite the clear visual enhancements, imperfect pseudo-ground-truth and inaccurate pose estimation limit more substantial quantitative gains over Vid2Avatar; see Sec. 4.3 for details. Cell color indicates best and second best for clearer comparisons.

| | ZJU-Mocap (Novel view) | | | | ZJU-Mocap (Novel pose) | | | | ZJU-Mocap (Shape) | | SynWild (Shape) | |
|---|---|---|---|---|---|---|---|---|---|---|---|---|
| | PSNR↑ | LPIPS↓ | FID↓ | KID↓ | PSNR↑ | LPIPS↓ | FID↓ | KID↓ | CD↓ | NC↑ | CD↓ | NC↑ |
| HumanNeRF | 29.66 | 36.78 | 28.35 | 14.23 | 29.57 | 34.17 | 42.84 | 12.32 | 0.051 | 0.765 | 0.507 | 0.635 |
| MonoHuman | 30.18 | 31.45 | 27.88 | 13.18 | 29.90 | 32.21 | 43.65 | 12.61 | 0.065 | 0.737 | N/A | N/A |
| NPC | 30.01 | 37.18 | 60.39 | 53.24 | 29.61 | 36.52 | 73.98 | 49.79 | 0.061 | 0.762 | 0.503 | 0.615 |
| Vid2Avatar | 29.76 | 35.61 | 36.83 | 27.65 | 29.53 | 35.69 | 54.16 | 31.51 | 0.042 | 0.852 | 0.499 | 0.687 |
| PM-Avatar | 30.24 | 38.38 | 49.58 | 39.64 | 29.87 | 39.26 | 65.71 | 40.16 | 0.051 | 0.766 | 0.500 | 0.632 |
| 3DGS-Avatar | 30.09 | 31.30 | 29.61 | 15.33 | 29.77 | 30.69 | 43.23 | 13.24 | 0.079 | 0.695 | 0.553 | 0.560 |
| GoMAvatar | 30.29 | 32.40 | 26.90 | 12.80 | 30.20 | 32.03 | 43.09 | 13.81 | 0.044 | 0.820 | 0.567 | 0.661 |
| Ours | 30.25 | 28.36 | 22.13 | 8.26 | 30.18 | 27.84 | 35.22 | 7.16 | 0.041 | 0.845 | 0.485 | 0.690 |

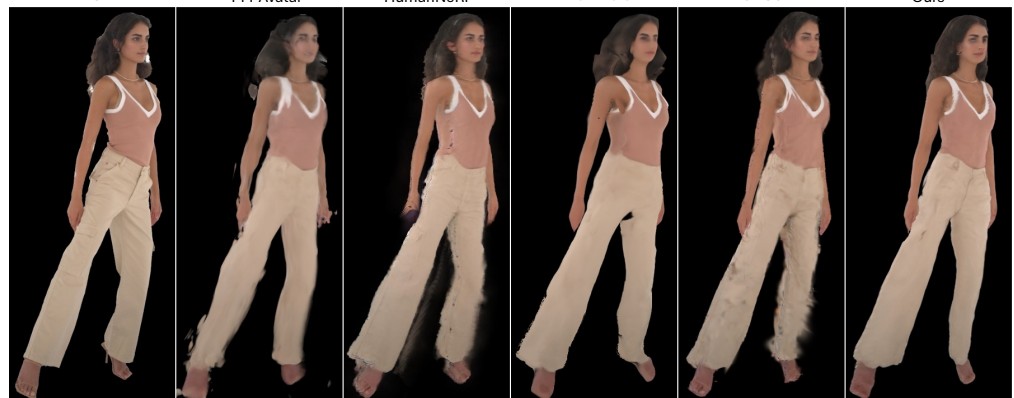

Figure 2: **Visual comparisons on the ActorsHQ dataset.** Our method achieves state-of-the-art results respecting geometry contours and texture generation on loose-fitting garments.

select four characters from ActorsHQ (Işık et al., 2023) for high-fidelity loose cloth modeling. We also adopt two sequences from MonoPerfCap (Xu et al., 2018) and downloaded Youtube videos (Weng et al., 2022; Yu et al., 2023) respectively as an in-the-wild dataset. The SynWild examples (Guo et al., 2023) are additionally applied to measure the capability of geometry reconstruction. Like PM-Avatar, we employ the open-source SAM model (Kirillov et al., 2023) to segment accurate foreground maps and obtain approximate camera and body poses with off-the-shelf estimators (Guo et al., 2023).

Following previous methods, we utilize pixel-wise Peak Signal-to-Noise Ratio (PSNR), perceptual metrics such as Learned Perceptual Image Patch Similarity (LPIPS) (Zhang et al., 2018), Fréchet Inception Distance (FID) (Heusel et al., 2017) and Kernel Inception Distance (KID) (Bińkowski et al., 2018) to assess image quality in pixel space and compare structural accuracy and textured details in latent semantic space. The metrics are computed across complete generation results, with LPIPS and KID scores are multiplied by 1000 for more clear demonstrations. To quantitatively estimate the generated shapes, we follow ARAH (Wang et al., 2022a) and PM-Avatar by computing Chamfer Distance (CD) and Normal Consistency (NC). In the following, we refer to "3DGS-Avatar" as "3DGS-A" for short.

## 4.2 RENDERING COMPARISONS

Starting with HumanNeRF, ZJU-Mocap has been widely used to test a method's ability to generalize to different camera views and novel poses. Given an image sequence, we use the provided first camera parameter for model training, and then use the remaining 22 cameras for evaluation (Weng et al., 2022; Yu et al., 2023; Wen et al., 2024).

In Fig. 4, compared to methods using pure part-related encodings including NPC and PM-Avatar, canonical space learning helps us provide more stable human shapes when driving with a novel pose.

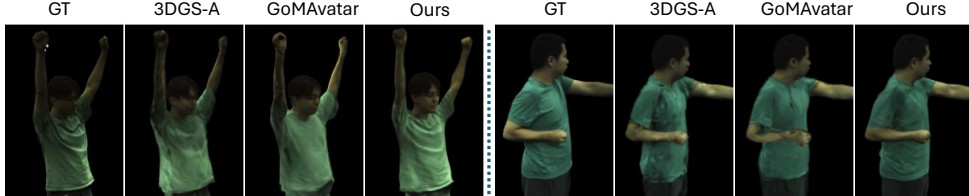

Figure 3: **Visual comparisons on ZJU-Mocap** with 3DGS based framework. Compared to the chosen baselines, our method can better preserve the sharp wrinkles (left column) without artifacts. The results for novel view synthesis and novel pose rendering are shown in both columns respectively.

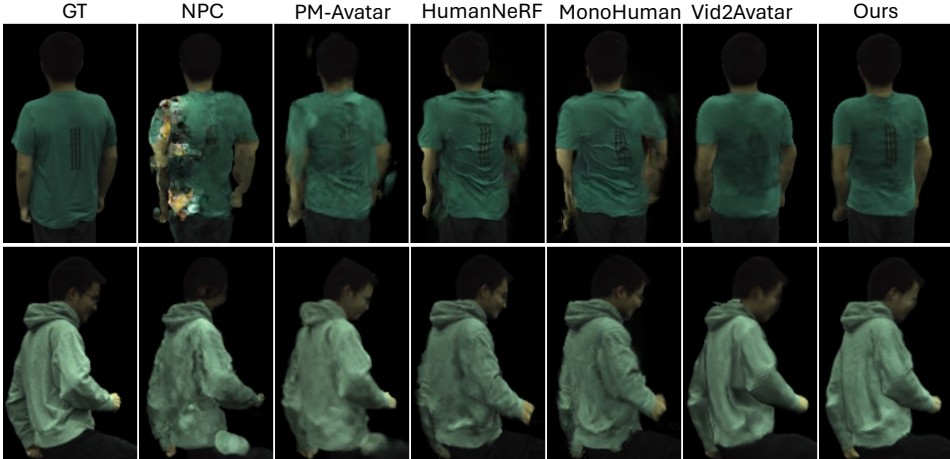

Figure 4: **Visual comparisons on ZJU-Mocap.** Our method can maintain large-scale body shapes and finely matched textures simultaneously while baselines either blur the notable patterns or greatly twist shape outlines under novel camera views (upper row) and novel avatar poses (bottom row).

Compared to methods explicitly modeling non-rigid motion, such as HumanNeRF and MonoHuman, our method significantly reduces pattern distortion and produces more reasonable body contours. It also markedly reduces blurry artifacts near the shape boundary. Compared to Vid2Avatar, which directly maps positions to SDF and color signals, our locality-sensitive avatars learn more fitting fine-grained textures. We can conclude similarly for 3DGS based methods in Fig. 3.

In Tab. 2, we calculate PSNR, LPIPS, KID and FID scores to quantitatively evaluate each method's image generation capability. Overall, our proposed approach produces comparable performance to best baselines on PSNR. However, from the table, we can see that our method's LPIPS, KID and FID scores significantly improve on all baselines, further supporting the advantages shown in Fig. 4 and Fig. 3. Note that the perceptual metrics (e.g. LPIPS) are reported to be more informative than the pixel-wise metrics like PSNR which are susceptible to slight output misalignment (Qian et al., 2024) and varying outdoor lighting conditions (Su et al., 2023).

Our advantages under novel views and novel poses extend to video sequences captured for high-fidelity loose cloths in Fig. 2. As methods rendering in observation space, both PM-Avatar and 3DGS-Avatar introduce blurry or floating artifacts near the pant boundaries, while HumanNeRF and Vid2Avatar distort the desired human shapes significantly. In contrast, our method maintains superior coarse contours with reasonably matched details which is numerically supported by Tab. 3 (a).

We further animate our model trained on a Youtube sequence with out-of-distribution body poses. Specifically, we collect poses of large-scale motions from AIST++ dataset (Li et al., 2021a) and compare in Fig. 5 to reveal our superior generalization to challenging animations.

### 4.3 COMPARISONS ON SHAPE RECONSTRUCTION

Fig. 6 shows the comparative results of 3D shape reconstruction on the ZJU-Mocap dataset. Our method can maintain consistent body boundaries and texture details closer to the ground truth. In contrast, the baselines either distort geometry with noisy artifacts or smooth out sharp textures, resulting in flat patterns (e.g., wrinkles).

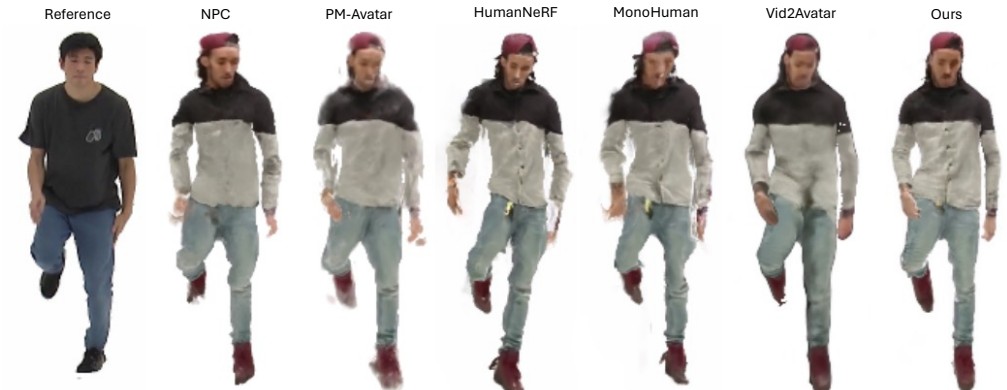

Figure 5: **Motion retargeting on Youtube video.** Given the out-of-distribution pose, only our method preserves accurate body contours with realistic details (e.g., cloth buttons), while baselines either smooth out important patterns or severely distort the human body.

Table 3: **(a) Quantitative comparisons on ActorsHQ dataset.** Being consistent with other evaluation testbeds in Tab. 2, our method significantly outperforms state-of-the-art baselines, especially in LPIPS. **(b) Ablation study on ZJU-Mocap sequence.** Our full model significantly outperforms all ablated baselines across all metrics, consistently proving the importance of all network components.

| | Novel view | | | Novel pose | | |
|---|---|---|---|---|---|---|
| | PSNR↑ | SSIM↑ | LPIPS↓ | PSNR↑ | SSIM↑ | LPIPS↓ |
| HumanNeRF | 21.57 | 0.855 | 171.91 | 21.42 | 0.853 | 172.68 |
| MonoHuman | 22.10 | 0.866 | 155.45 | 22.03 | 0.863 | 155.53 |
| PM-Avatar | 19.59 | 0.791 | 212.21 | 19.39 | 0.792 | 209.97 |
| Vid2Avatar | 20.32 | 0.909 | 136.67 | 20.46 | 0.910 | 133.65 |
| 3DGS-Avatar | 22.36 | 0.921 | 123.70 | 22.22 | 0.921 | 123.06 |
| GauHuman | **23.53** | 0.926 | 126.74 | **23.68** | 0.927 | 124.11 |
| **Ours** | 22.97 | **0.930** | **103.78** | 23.10 | **0.933** | **100.95** |

(a)

| | Novel view | | | Novel pose | | |
|---|---|---|---|---|---|---|
| | PSNR↑ | SSIM↑ | LPIPS↓ | PSNR↑ | SSIM↑ | LPIPS↓ |
| onlyPose | 33.04 | 0.977 | 27.39 | 33.20 | 0.976 | 27.57 |
| w/o locality | 32.48 | 0.974 | 28.95 | 32.86 | 0.975 | 28.02 |
| w/o canonical | 33.08 | 0.977 | 27.92 | 32.81 | 0.975 | 29.78 |
| w/o GNN | 32.75 | 0.975 | 27.47 | 32.83 | 0.975 | 27.88 |
| w/o window | 32.91 | 0.976 | 27.65 | 33.07 | 0.975 | 27.47 |
| w/o $\mathcal{L}_{\triangle x}$ | 33.16 | 0.978 | 26.14 | 33.13 | 0.976 | 27.50 |
| **Ours** | **33.36** | **0.978** | **25.58** | **33.44** | **0.977** | **26.34** |

(b)

Since Vid2Avatar and GoMAvatar also aim for feasible geometry reconstruction, we further highlight our advantages in generating geometry details by presenting the normal map results in Fig. A. In comparison with baselines, our method can more faithfully synthesize the facial structure in both avatars and the zipper pattern on the left. Additionally, we can show more realistic textured variations (e.g. wrinkles) in the whole human body in both examples, featuring our benefits in outputting high-frequency geometric patterns.

We also follow ARAH (Wang et al., 2022a) to calculate pseudo ground truth and report scores in the "ZJU-Mocap (Shape)" column of Tab. 2 to quantitatively support our conclusions. Note that our Normal Consistency (NC) scores are similar to Vid2Avatar's because the pseudo ground truth used by ARAH smooths out many desired surface details and leads to imperfect evaluations. Please refer to Sec. D and Fig. O for more discussions.

As suggested by Vid2Avatar, we use the semi-synthetic SynWild Dataset as the evaluation protocal for monocular human surface reconstruction and list the quantitative metrics in the "SynWild" column of Tab. 2. Despite the clear visual enhancements in Fig. 7, inaccurate human pose estimation limits more substantial quantitative gains over Vid2Avatar. How to better measure the quality of generated shapes remains an open problem. Also despite our best efforts in picking representative images in front and back views, we omit listing metrics for MonoHuman due to model training divergence. We list the per-object scores in Tab. I and Tab. J for the ZJU-Mocap and SynWild datasets respectively.

## 4.4 ABLATION STUDIES

We study the effect of various important network components on the ZJU-MoCap S386 sequence. Specifically, we perform the following models: 1) Removing the canonical position $x_c$ as w/o canonical; 2) Only preserving the skeletal deformation as w/o locality; 3) Removing the window function as w/o window; 4) Removing the GNN but use $x_c$ to regress $\triangle x$ as w/o GNN; 5) Removing the regularization loss on non-rigid offset $\triangle x$ as w/o $\mathcal{L}_{\triangle x}$. It shows in Fig. 8 that only utilizing locality information as w/o canonical cannot reconstruct a full human body. The w/o window model smashes the left hand when two parts are close and cause entangled features.

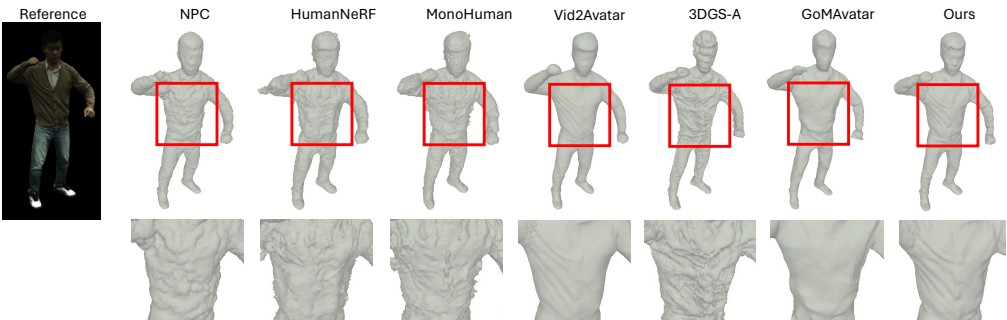

Figure 6: **Geometry reconstruction on ZJU-Mocap.** Our method can generate sharper geometric structures and adaptive fine-grained details than all baselines. In comparison, the baselines either significantly distort geometric patterns with noise or weaken notable cloth wrinkles.

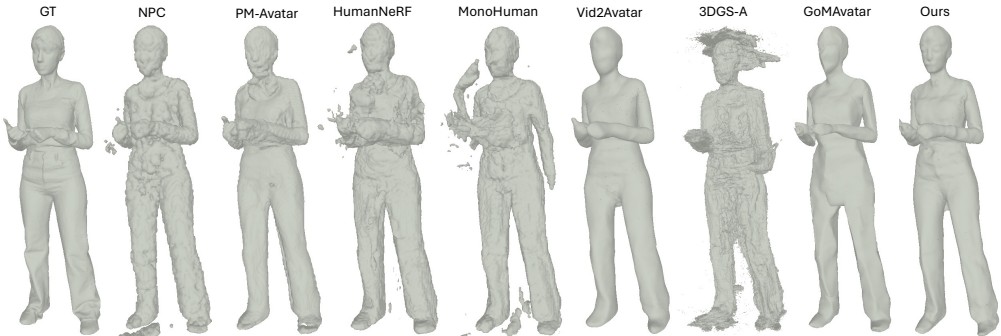

Figure 7: **Geometry reconstruction on SynWild.** Our method improves existing baselines by preserving superior coarse outlines together with finely matched geometric details.

With canonical representation, both w/o locality and w/o GNN models are capable of reproducing structured patterns (e.g. the vertical strips in the back area) but seriously distort local patterns and overfit specific wrinkles. Without offset regularization, the w/o $\mathcal{L}_{\Delta \mathrm{x}}$ model fails to preserve the constant left arm with weird distortions. In contrast, our full model can synthesize continuous boundary with detailed vertical marks. The quantitative metrics are further reported in Tab. 3 (b). We show that all proposed techniques are required to reach the optimal performance, best reflected by LPIPS which is the most informative metric for generalization evaluation under a monocular setup.

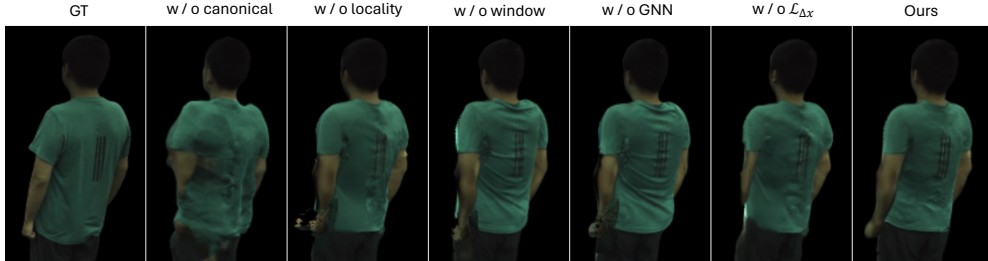

Figure 8: **Ablation Study**. Only the full model with all components can faithfully synthesize the structured patterns (e.g. the strip textures) and avoid artifacts and contour distortions.

## 5 CONCLUSIONS

We present a novel locality-sensitive avatar representation for monocular videos. The key innovation is utilizing part relationships captured by a graph neural network to estimate 3D offsets as non-rigid motion, enabling the faithful reproduction of locally adaptive details. Our method outperforms the existing baselines by estimating visually pleasing geometry and achieving photo-realistic rendering, where both coarse body contours and pose-dependent cloth wrinkles can be simultaneously captured.

ACKNOWLEDGEMENT

This work was supported by the Natural Sciences and Engineering Research Council of Canada (NSERC) Discovery Grant and by Advanced Research Computing at the University of British Columbia. This work was also funded, in part, by the Vector Institute for AI, Canada CIFAR AI Chairs, NSERC Canada Research Chair (CRC). It was also partially supported by the Wallenberg AI, Autonomous Systems and Software Program (WASP) funded by the Knut and Alice Wallenberg Foundation, Sweden. We thank ARC at UBC and Compute Canada for providing computational resources. We also thank the anonymous reviewers for the helpful discussions.

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

# Appendices

In the appendix, we first provide details of our method implementation and evaluated datasets. Then we append illustration examples to support our technical motivations. Additionally, we present the visual outputs of both rigid deformation and full model, followed by detailed quantitative comparisons across different datasets and different metrics. Following HumanNeRF (Weng et al., 2022) and Vid2Avatar (Guo et al., 2023), we show the appearance of our reconstructed 3D avatars in canonical space. Finally, we discuss the limitations and social impacts of this project. See the attached video for the animation results.

## A  IMPLEMENTATION DETAILS

To preserve the comparison consistency for better model generalization evaluations, we maintain the same hyper-parameter settings across all experiments, which include the weights of loss function $\{\lambda_{\text{eik}}, \lambda_{\text{LPIPS}}, \lambda_{\Delta\text{x}}\}$, the number of training iterations, and the network capacity and learning rate. We choose the hyper-parameters depending on empirical performance on used benchmarks. Specifically, our method implementation is based on PyTorch framework (Paszke et al., 2019). Similar to HumanNeRF and PM-Avatar, we utilize the Adam optimizer (Kingma & Ba, 2014) to optimize our loss function $\mathcal{L}$ with default settings $\beta_1 = 0.9$ and $\beta_2 = 0.99$. We set the initial learning rate of the learnable parameter $\beta$ to $1 \times 10^{-4}$ for stable training and the learning rates of remaining parameters to $5 \times 10^{-4}$. The step decay schedule is applied to adjust the learning rate for better network convergence. Like PM-Avatar, we set $\lambda_{\text{s}} = 0.001$ and $N_B = 24$ to accurately capture the topology variations and avoid introducing unnecessary training changes. All learnable weights are activated by Relu (Agarap, 2018) for network stability. During training, the initially estimated human poses from offline public models are refined to reduce pose estimation errors. Additionally, we choose the VGG (Simonyan & Zisserman, 2014) network as the backbone of our LPIPS objective. To more effectively minimize our loss function, we sample patches with size $H \times H$ on an input image and produce rendered patch with volume rendering process rather than training on random ray samples. Both the rendered patches and the corresponding ground truth patch are embedded into VGG feature space to compute LPIPS metrics. Here we sample 4 patches with $H = 24$. Similar to HumanNeRF, we employ the delayed optimization strategy for $\Delta x$ to mitigate the overfitting of the non-rigid motions to input images. Specifically, we disable the non-rigid motions at the beginning of network training, and then bring them back after 5000 iterations. We train our network on two NVIDIA Tesla V100 GPUs for 15 hours.

## B  MORE DETAILS ABOUT DATASETS

**ZJU-Mocap dataset (Peng et al., 2021b).** Following the most recent state-of-the-art baselines (Qian et al., 2024; Lei et al., 2024; Wen et al., 2024), we leverage ZJU-Mocap as the major testbed for evaluation. Specifically, we pick eight sequences (S313, S377, S386, S387, S390, S392, S393, S394) from the ZJU-MoCap dataset and follow the data split strategy of MonoHuman (Yu et al., 2023) and

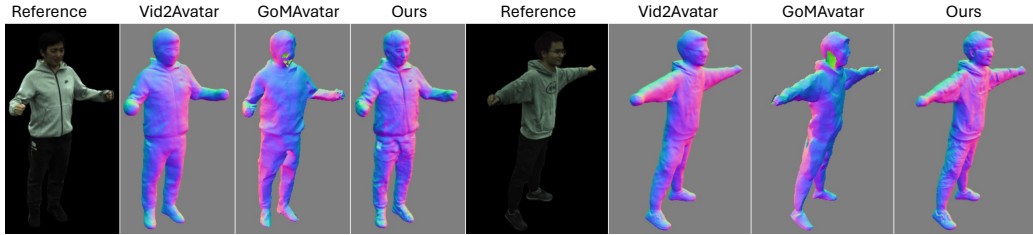

Figure A: **Qualitative Comparisons for Normal Map.** Compared to baselines, our method can generate more significant textures with faithful details, such as the avatar facial structures and the cloth textures (e.g. the zipper patter on the left) in both examples.

Table A: **Novel-view synthesis comparisons on ZJU-Mocap (Peng et al., 2021b)** with PSNR and LPIPS scores. Cell color indicates best and second best . Although our method achieves comparable PSNR than GoMAvatar, it consistently outperforms the best baseline in LPIPS with over **10%** improvement. Note that LPIPS is reported to be more informative to evaluate the output quality due to unavoidable slight misalignments in the monocular settings (Qian et al., 2024).

| | HumanNeRF | | MonoHuman | | NPC | | Vid2Avatar | | PM-Avatar | | 3DGS-Avatar | | GoMAvatar | | **Ours** | |
|---|---|---|---|---|---|---|---|---|---|---|---|---|---|---|---|---|
| | PSNR↑ | LPIPS↓ | PSNR↑ | LPIPS↓ | PSNR↑ | LPIPS↓ | PSNR↑ | LPIPS↓ | PSNR↑ | LPIPS↓ | PSNR↑ | LPIPS↓ | PSNR↑ | LPIPS↓ | PSNR↑ | LPIPS↓ |
| S313 | 29.49 | 30.46 | 29.55 | 31.40 | 29.70 | 36.29 | 28.44 | 37.74 | 29.93 | 38.51 | 30.01 | 28.90 | 29.41 | 28.61 | 28.65 | 28.74 |
| S377 | 29.79 | 28.49 | 30.46 | 20.91 | 30.52 | 22.29 | 29.52 | 24.91 | 30.66 | 24.47 | 30.17 | 21.37 | 30.60 | 23.91 | 30.36 | 18.34 |
| S386 | 32.10 | 41.84 | 32.99 | 30.97 | 32.07 | 36.88 | 33.04 | 30.65 | 32.54 | 34.66 | 32.64 | 31.32 | 32.97 | 30.36 | 33.36 | 25.58 |
| S387 | 28.11 | 37.46 | 28.40 | 35.06 | 27.76 | 43.34 | 28.15 | 40.26 | 28.14 | 44.48 | 28.09 | 34.18 | 28.34 | 36.30 | 28.49 | 32.53 |
| S390 | 30.16 | 33.09 | 30.27 | 34.68 | 29.74 | 46.02 | 30.73 | 37.33 | 30.28 | 42.51 | 30.20 | 35.82 | 30.73 | 35.42 | 30.85 | 31.58 |
| S392 | 30.20 | 40.06 | 30.98 | 30.80 | 31.70 | 33.34 | 30.46 | 36.58 | 31.47 | 35.86 | 31.07 | 30.84 | 31.04 | 33.25 | 31.31 | 28.04 |
| S393 | 28.16 | 40.85 | 28.54 | 34.97 | 28.61 | 40.88 | 27.94 | 40.93 | 28.85 | 43.09 | 28.75 | 35.10 | 28.80 | 37.77 | 28.59 | 32.49 |
| S394 | 29.28 | 41.97 | 30.21 | 32.80 | 30.07 | 38.38 | 29.82 | 36.46 | 30.30 | 43.43 | 29.77 | 32.88 | 30.44 | 33.56 | 30.38 | 29.61 |
| Avg | 29.66 | 36.78 | 30.18 | 31.45 | 30.01 | 37.18 | 29.76 | 35.61 | 30.27 | 38.38 | 30.09 | 31.30 | 30.29 | 32.40 | 30.25 | 28.36 |

Table B: **Novel-view synthesis comparisons on ZJU-Mocap (Peng et al., 2021b)** with FID and KID metrics. Cell color indicates best and second best . The locality sensitive avatar representation enables over **20%** improvement in FID and over **55%** improvement in KID than the best baselines.

| | HumanNeRF | | MonoHuman | | NPC | | Vid2Avatar | | PM-Avatar | | 3DGS-Avatar | | GoMAvatar | | **Ours** | |
|---|---|---|---|---|---|---|---|---|---|---|---|---|---|---|---|---|
| | FID↓ | KID↓ | FID↓ | KID↓ | FID↓ | KID↓ | FID↓ | KID↓ | FID↓ | KID↓ | FID↓ | KID↓ | FID↓ | KID↓ | FID↓ | KID↓ |
| S313 | 20.06 | 9.43 | 18.88 | 7.25 | 58.09 | 54.37 | 36.07 | 30.08 | 51.49 | 45.93 | 24.26 | 14.06 | 20.90 | 10.63 | 16.87 | 6.22 |
| S377 | 16.93 | 5.37 | 15.90 | 3.95 | 15.84 | 4.75 | 20.78 | 12.52 | 18.47 | 8.02 | 13.90 | 3.61 | 18.53 | 7.84 | 11.72 | 1.58 |
| S386 | 47.39 | 31.06 | 46.28 | 28.08 | 85.00 | 79.70 | 37.44 | 23.42 | 49.77 | 34.13 | 43.47 | 23.80 | 33.34 | 16.83 | 26.23 | 9.12 |
| S387 | 34.07 | 15.70 | 39.43 | 21.99 | 88.20 | 80.01 | 41.46 | 25.17 | 73.66 | 63.36 | 37.69 | 20.15 | 36.73 | 18.44 | 30.59 | 10.85 |
| S390 | 30.81 | 19.53 | 30.78 | 20.00 | 73.95 | 71.72 | 31.81 | 22.27 | 57.71 | 54.59 | 40.64 | 28.27 | 31.18 | 17.90 | 25.48 | 14.19 |
| S392 | 24.35 | 9.88 | 21.41 | 6.26 | 45.40 | 38.44 | 38.54 | 31.50 | 42.99 | 34.49 | 23.43 | 9.57 | 23.86 | 10.73 | 20.71 | 8.21 |
| S393 | 25.30 | 12.67 | 24.42 | 10.20 | 60.75 | 55.83 | 43.53 | 39.11 | 52.15 | 40.33 | 26.06 | 12.65 | 24.44 | 11.27 | 19.91 | 6.71 |
| S394 | 27.91 | 10.18 | 25.92 | 7.64 | 55.92 | 41.14 | 44.99 | 37.11 | 50.37 | 36.26 | 27.43 | 10.51 | 26.18 | 8.72 | 25.53 | 9.22 |
| Avg | 28.35 | 14.23 | 27.88 | 13.18 | 60.39 | 53.24 | 36.83 | 27.65 | 49.58 | 39.64 | 29.61 | 15.33 | 26.90 | 12.80 | 22.13 | 8.26 |

GoMAvatar (Wen et al., 2024). We use the images captured by the first camera for training and the remaining images from other 22 cameras for evaluation.

**Youtube Online Videos.** Starting from HumanNeRF (Weng et al., 2022), online Youtube sequences are widely used to assess the generalization capability to in-the-wild monocular videos. In this paper, we downloaded "Story" and "Invisible" videos from Internet. Since these videos are only captured by a monocular camera and contains diverse human motions, we quantitatively measure our method and other baselines in the task of novel pose animation.

**MonoPerfCap Dataset (Xu et al., 2018).** This dataset contains in-the-wild human performance sequences with the complex real-world environment and distinctive human actions. It also provide high-resolution images with ground-truth masks. Like NPC (Su et al., 2023) and PM-Avatar (Song et al., 2024a), we also apply the MonoPerfCap dataset to evaluate the robustness to unseen poses in monocular videos.

**SynWild Dataset (Guo et al., 2023).** Accurate geometry reconstruction from monocular videos is a core aspect of our method. In the SynWild dataset, dynamic human subjects are captured using a multi-view system, allowing for detailed geometry and texture reconstruction via commercial software. These reconstructed shapes serve as semi-synthetic meshes, enabling approximate yet realistic quantitative comparisons.

**ActorHQ Dataset (Işık et al., 2023).** We choose three characters from the dataset for our experiments, including one female with loose dresses, one female with loose-fitting pants and one male character with tighter clothes. For each sequence, we use the images captured by 'camera 127' and 'camera 128' as training and evaluation data respectively as they can cover the whole human body. Specifically, we pick one image every four frames until we have 375 images in total for training and use 125 images and 175 images for the evaluation of novel pose synthesis and novel view rendering respectively.

## C  MOTIVATION CLARIFICATION

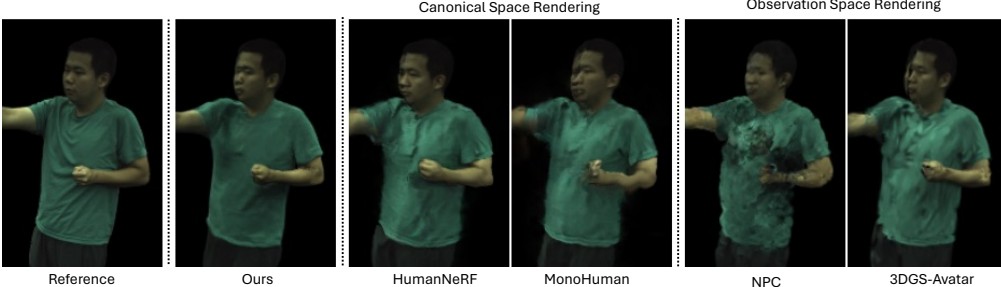

Figure B: **Motivation demonstration.** In recent literature of avatar modeling, the deformation to canonical space is learned to facilitate generalizations to different poses while the spatial locality encoding aims to produce fine-grained patterns. However, none of these two strategies on their own suffices to provide a visually good rendering. Specifically, the deformation based on pure global coordinates (e.g. HumanNeRF and MonoHuman) twists the local human structures (e.g. arm and face contours) while the part-wise relationship without canonical representation learning (e.g. NPC) introduces unseen black artifacts in the torso area. By enforcing the non-rigid deformation to be locally sensitive, our method can faithfully preserve both smooth patterns and the structured facial appearance and yield best rendered result.

Fig. B visualizes our motivation discussions mentioned in Sec. 1. While two different parts expect distinct texture changes during the avatar deformation, the previous locality encoding methods (e.g. NPC) introduce unseen black artifacts when rendering under novel human poses. Meanwhile, methods with monolithic motion modeling (e.g. HumanNeRF and MonoHuman) are non-adaptive, and hence significantly twists shape contours with distorted artifacts. The recent 3DGS based method (3DGS-Avatar), which transforms a set of template Gaussians to a per-frame observation space for fast rendering, also cannot synthesize photorealistic outputs. While the representation in canonical space is well solved with NeRFs, SDFs, or Gaussians, the key challenge remains to keep pose-independent information and adaptive non-linear details simultaneously which motivates our method.

Inspired by previous methods, we leverage GNN as a simple way to process human skeletons and extract local features. Compared to other options to extract local features, such as time-dependent voxel planes (Wu et al., 2024), the GNN based feature extraction paradigm has the following benefits: (1) GNN propagates features within neighboring regions and can adaptively capture local patterns in multiple scales; (2) As inputs to GNN, the human skeleton is sparse and is able to yield decent expressiveness with limited computation complexity; (3) Pose estimation models can also work for testing poses and enable out-of-distribution pose animation.

In comparison, time-dependent voxel planes might struggle to model complex local relationships and cannot extend to novel pose animation due to fixed space-time resolutions. They are also dense in memory storage, which limits their capability in processing fine-grained details. The effectiveness of GNN can be revealed via our ablation study results and comparisons with SOTA methods.

## D  MORE RESULTS

**Network Outputs.** As shown in Fig. C, the output RGB image and normal map from the rigid deformation smooth out necessary high-frequency patterns while our full model with localized non-rigid offsets successfully supplements the fine-grained details. The visual difference clearly advocates our conceptual motivation that localized non-rigid deformations can promote adaptive visual detail synthesis.

**Canonical Space Visualizations.** An expressive canonical space of avatar representations is crucial for deformation-based human modeling. Therefore, we selected five representative examples from the ZJU-MoCap dataset and visualized the front and back views of our reconstructed 3D avatars in the canonical space, as shown in Fig. K. Our canonical images successfully capture both the large-scale outline patterns and fine-grained texture details.

Table C: **Novel-pose synthesis comparisons on ZJU-Mocap (Peng et al., 2021b)** with PSNR and LPIPS metrics. Cell color indicates best and second best . Compared to PSNR, LPIPS is more informative to evaluate the output quality due to the monocular setting (Qian et al., 2024). Overall, our proposed approach produces comparable performance to the best baseline (GoMAvatar) in PSNR but significantly surpasses all the baselines on LPIPS with over **10%** improvement.

| | HumanNeRF | | MonoHuman | | NPC | | Vid2Avatar | | PM-Avatar | | 3DGS-Avatar | | GoMAvatar | | **Ours** | |
|---|---|---|---|---|---|---|---|---|---|---|---|---|---|---|---|---|
| | PSNR ↑ | LPIPS ↓ | PSNR ↑ | LPIPS ↓ | PSNR ↑ | LPIPS ↓ | PSNR ↑ | LPIPS ↓ | PSNR ↑ | LPIPS ↓ | PSNR ↑ | LPIPS ↓ | PSNR ↑ | LPIPS ↓ | PSNR ↑ | LPIPS ↓ |
| S313 | 28.08 | 32.37 | 28.24 | 32.81 | 27.41 | 37.13 | 28.00 | 38.43 | 28.12 | 40.22 | 28.47 | 29.53 | 28.49 | 29.34 | 28.45 | 27.61 |
| S377 | 29.91 | 23.87 | 30.77 | 21.67 | 30.61 | 21.55 | 29.80 | 24.77 | 30.91 | 24.00 | 29.89 | 21.42 | 30.68 | 23.41 | 30.86 | 17.38 |
| S386 | 32.62 | 39.36 | 32.97 | 32.73 | 31.71 | 37.95 | 32.48 | 32.43 | 32.46 | 36.39 | 32.43 | 31.85 | 32.86 | 32.25 | 33.44 | 26.34 |
| S387 | 28.01 | 35.27 | 27.93 | 33.45 | 27.24 | 42.14 | 27.34 | 40.83 | 27.67 | 44.11 | 27.83 | 33.53 | 28.18 | 36.43 | 27.89 | 31.46 |
| S390 | 30.01 | 32.24 | 30.62 | 35.84 | 30.02 | 42.60 | 30.81 | 34.73 | 30.18 | 44.47 | 30.59 | 33.10 | 31.10 | 32.59 | 31.21 | 28.57 |
| S392 | 30.95 | 34.23 | 31.24 | 31.04 | 31.33 | 34.88 | 30.50 | 37.53 | 31.10 | 38.20 | 31.07 | 30.69 | 31.44 | 33.20 | 31.27 | 29.01 |
| S393 | 28.43 | 36.26 | 28.46 | 34.24 | 28.76 | 39.03 | 28.09 | 40.17 | 28.92 | 43.44 | 28.64 | 32.97 | 29.09 | 36.02 | 28.81 | 32.33 |
| S394 | 28.52 | 39.75 | 28.94 | 35.90 | 29.77 | 36.90 | 29.24 | 36.61 | 29.56 | 43.21 | 29.27 | 32.45 | 29.79 | 33.00 | 29.47 | 30.02 |
| Avg | 29.57 | 34.17 | 29.90 | 32.21 | 29.61 | 36.52 | 29.53 | 35.69 | 29.87 | 39.26 | 29.77 | 30.69 | 30.20 | 32.03 | 30.18 | 27.84 |

Table D: **Novel-pose synthesis comparisons on ZJU-Mocap (Peng et al., 2021b)** with FID and KID. Cell color indicates best and second best . Our developed algorithm yields over **20%** improvement in FID and over **70%** improvement in KID than the top baselines, showing better generalization to unseen poses.

| | HumanNeRF | | MonoHuman | | NPC | | Vid2Avatar | | PM-Avatar | | 3DGS-Avatar | | GoMAvatar | | **Ours** | |
|---|---|---|---|---|---|---|---|---|---|---|---|---|---|---|---|---|
| | FID ↓ | KID ↓ | FID ↓ | KID ↓ | FID ↓ | KID ↓ | FID ↓ | KID ↓ | FID ↓ | KID ↓ | FID ↓ | KID ↓ | FID ↓ | KID ↓ | FID ↓ | KID ↓ |
| S313 | 35.17 | 12.17 | 30.94 | 8.02 | 69.17 | 53.82 | 47.84 | 32.55 | 61.29 | 45.30 | 36.56 | 15.18 | 35.28 | 13.72 | 25.75 | 4.98 |
| S377 | 30.84 | 5.63 | 29.67 | 3.31 | 29.43 | 4.42 | 36.44 | 15.94 | 31.77 | 7.92 | 26.22 | 3.04 | 32.18 | 9.11 | 23.81 | 1.41 |
| S386 | 58.84 | 25.23 | 62.04 | 25.10 | 93.65 | 68.22 | 51.66 | 24.64 | 65.01 | 31.79 | 53.92 | 18.28 | 50.36 | 16.26 | 36.83 | 6.76 |
| S387 | 59.92 | 17.47 | 65.40 | 23.90 | 116.49 | 83.09 | 66.61 | 29.66 | 97.15 | 63.34 | 60.80 | 21.69 | 57.04 | 18.54 | 49.46 | 8.33 |
| S390 | 37.75 | 12.43 | 42.95 | 17.55 | 81.35 | 60.21 | 48.50 | 28.09 | 77.29 | 61.10 | 50.52 | 23.35 | 47.25 | 23.06 | 36.11 | 12.42 |
| S392 | 38.32 | 5.93 | 37.74 | 5.09 | 66.68 | 44.54 | 64.31 | 46.91 | 61.38 | 36.81 | 36.82 | 7.87 | 40.13 | 9.22 | 38.00 | 10.16 |
| S393 | 40.47 | 10.70 | 39.91 | 9.47 | 74.95 | 51.61 | 59.17 | 38.36 | 67.32 | 39.20 | 39.34 | 7.84 | 41.67 | 11.55 | 34.66 | 6.05 |
| S394 | 41.39 | 8.95 | 40.56 | 8.41 | 60.10 | 32.45 | 58.75 | 35.94 | 64.46 | 35.82 | 41.67 | 8.63 | 40.82 | 8.99 | 37.11 | 7.14 |
| Avg | 42.84 | 12.32 | 43.65 | 12.61 | 73.98 | 49.79 | 54.16 | 31.51 | 65.71 | 40.16 | 43.23 | 13.24 | 43.09 | 13.81 | 35.22 | 7.16 |

**Normal Map Comparisons.** Following former avatar modeling papers Wang et al. (2022a); Guo et al. (2023), we visualize the produced normal maps to qualitatively compare geometry reconstruction in Fig. A. As most latest baseline, GoMAvatar produces less detailed and even physically implausible reconstructions (e.g. incorrect normal directions and discontinuous shape contours). While Vid2Avatar can preserve reasonable body outlines, it oversmoothes the sharp patterns (e.g. human face structures) and blurs desired high-frequency details (e.g. wrinkles). In contrast, our method successfully reproduces both coarse human shapes and fine-grained signals.

**Detailed Quantative Comparisons.** Here we further provide detailed score comparisons for image renderings and shape reconstructions. Specifically, Tab. A and Tab. B list the PSNR, LPIPS, FID and KID scores for the novel view synthesis on ZJU-Mocap. Correspondingly, Tab. C and Tab. D present the per-sequence results for novel pose animation on the same dataset, using the same evaluation metrics. Additionally, Tab. E and Tab. F provide the evaluation results for the four ActorsHQ examples, reporting PSNR, SSIM, and LPIPS scores. Finally, Tab. I and Tab. J look into geometry reconstruction comparisons on ZJU-Mocap and SynWild datasets respectively with Chamfer Distance (CD) and Normal Consistency (NC).

As discussed in Sec. 3.3 of the MonoHuman paper, front and back view images must be selected for each sequence before training. Despite our best efforts to correctly select these views for the S1 sequence, the model diverged from the start, leading us to exclude MonoHuman when calculating the mean values. Nevertheless, our method consistently outperforms all baselines in quantitative results on the remaining sequences.

**More Visual Comparisons.** Besides the image results shown in the main text, we visualize more rendering comparisons for ZJU-Mocap in Fig. D and Fig. E, corresponding geometry reconstruction in Fig. I. Consistent with the findings in the main text, our method maintains coarse body shapes and finely matched textures simultaneously for both novel view synthesis and unseen pose animation. To

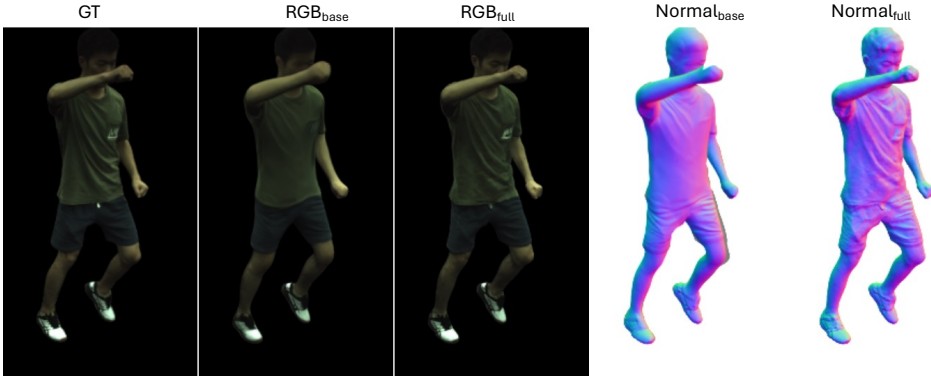

| GT | RGB$_{base}$ | RGB$_{full}$ | Normal$_{base}$ | Normal$_{full}$ |

Figure C: **Outputs from the rigid deformation and full model.** Adhering to our network design, the localized offsets as the non-rigid motion are learned to carve the high-frequency details for the features from holistic rigid deformation. Thus, compared to the RGB rendering (RGB$_{base}$) and normal map (Normal$_{base}$) from the pure rigid deformation which present overall smooth patterns, our full model can produce more fine-grained patterns in both geometric output (Normal$_{full}$) and visual image (RGB$_{full}$).

Table E: **Comparison with methods rendering in canonical space on the ActorsHQ videos.** Our method demonstrates superior synthesis quality in learning loose clothing from monocular videos.

| | HumanNeRF | | | MonoHuman | | | Vid2Avatar | | | **Ours** | | |
|---|---|---|---|---|---|---|---|---|---|---|---|---|
| | PSNR↑ | SSIM↑ | LPIPS↓ | PSNR↑ | SSIM↑ | LPIPS↓ | PSNR↑ | SSIM↑ | LPIPS↓ | PSNR↑ | SSIM↑ | LPIPS↓ |
| **Novel View Synthesis** | | | | | | | | | | | | |
| Actor01 | 23.29 | 0.894 | 135.19 | 23.98 | 0.911 | 125.54 | 20.54 | 0.916 | 121.78 | 24.0 | 0.944 | 86.59 |
| Actor03 | 18.59 | 0.723 | 290.59 | 18.52 | 0.707 | 252.01 | 20.14 | 0.890 | 165.89 | 20.89 | 0.899 | 142.39 |
| Actor06 | 22.46 | 0.905 | 113.46 | 21.70 | 0.925 | 114.42 | 19.98 | 0.919 | 116.28 | 22.78 | 0.942 | 82.06 |
| Actor07 | 21.96 | 0.898 | 148.41 | 24.19 | 0.921 | 129.83 | 20.63 | 0.912 | 142.75 | 24.21 | 0.937 | 104.10 |
| **Novel Pose Rendering** | | | | | | | | | | | | |
| Actor01 | 22.78 | 0.890 | 141.31 | 23.57 | 0.907 | 130.24 | 20.41 | 0.912 | 124.96 | 23.31 | 0.938 | 92.98 |
| Actor03 | 18.42 | 0.711 | 293.30 | 18.52 | 0.699 | 250.46 | 20.59 | 0.897 | 157.64 | 22.20 | 0.915 | 126.21 |
| Actor06 | 22.03 | 0.908 | 111.91 | 21.51 | 0.925 | 113.91 | 20.06 | 0.920 | 113.23 | 22.69 | 0.941 | 81.33 |
| Actor07 | 22.46 | 0.903 | 144.20 | 24.50 | 0.922 | 127.51 | 20.79 | 0.913 | 138.77 | 24.22 | 0.937 | 103.29 |

better recognize our visual improvements in the demo video, we pick six consecutive frames of the ZJU-Mocap sequence and present results in Fig. J. It is clear that our method successfully preserves realistic outputs across different frames while baselines generate various artifacts. In Fig. N, we additionally compare with Vid2Avatar on the geometry reconstruction under the setting of ActorsHQ dataset. As illustrated in the close-ups, our method yields much more clear facial structures and more detailed pant outlines while Vid2Avatar simply smooths out all necessary delicate details.

**Results on outdoor datasets.** Following PM-Avatar and HumanNeRF, we provide extra results on sequences selected from MonoPerfCap and YouTube videos. In Fig. L, our method generates the most realistic shape outlines and the most reasonable and adaptive texture details. In contrast, the baselines either significantly distort foreground parts (e.g., arms on $2^{nd}$ row) or blur the texture folds on clothing. Correspondingly, in Tab. H, we report LPIPS, FID and KID scores to quantitatively evaluate each method's capability in generating detailed textures. Consistent with the improvements on ZJU-Mocap sequences, our method reveals superior capabilities in structured texture synthesis on in-the-wild datasets compared to baselines by showing better quantitative metrics.

**Results on MVHumanNet dataset.** Besides the four sequences chosen in the ActorsHQ dataset, we additionally evaluate on the latest released MVHumanNet dataset (Xiong et al., 2024) to display our generalization to complex body-cloth interactions. Specifically, we select three sequences with varying degrees of clothing looseness and compare our approach against two NeRF-based methods and three 3DGS-based methods. The qualitative and quantitative results are shown in Fig. F- H and

Table F: **Comparison with methods rendering in observation space on the ActorsHQ videos.** Our method demonstrates superior synthesis quality in learning loose clothing from monocular videos.

| | 3DGS-Avatar | | | GauHuman | | | PM-Avatar | | | **Ours** | | |
|---|---|---|---|---|---|---|---|---|---|---|---|---|
| | PSNR↑ | SSIM↑ | LPIPS↓ | PSNR↑ | SSIM↑ | LPIPS↓ | PSNR↑ | SSIM↑ | LPIPS↓ | PSNR↑ | SSIM↑ | LPIPS↓ |
| **Novel View Synthesis** | | | | | | | | | | | | |
| Actor01 | 22.74 | 0.932 | 101.75 | 24.16 | 0.931 | 116.98 | 21.80 | 0.917 | 139.42 | 24.0 | 0.944 | 86.59 |
| Actor03 | 20.57 | 0.890 | 175.95 | 21.94 | 0.900 | 160.48 | 8.99 | 0.370 | 491.67 | 20.89 | 0.899 | 142.39 |
| Actor06 | 22.08 | 0.930 | 102.10 | 22.82 | 0.933 | 104.05 | 22.52 | 0.936 | 102.02 | 22.78 | 0.942 | 82.06 |
| Actor07 | 24.06 | 0.935 | 115.00 | 25.19 | 0.938 | 125.44 | 25.08 | 0.943 | 115.71 | 24.21 | 0.937 | 104.10 |
| **Novel Pose Rendering** | | | | | | | | | | | | |
| Actor01 | 22.19 | 0.927 | 107.45 | 23.84 | 0.927 | 121.12 | 21.53 | 0.913 | 143.71 | 23.31 | 0.938 | 92.98 |
| Actor03 | 20.53 | 0.889 | 173.71 | 23.08 | 0.910 | 150.82 | 8.72 | 0.379 | 483.45 | 22.20 | 0.915 | 126.21 |
| Actor06 | 22.09 | 0.932 | 99.16 | 22.74 | 0.933 | 102.45 | 22.35 | 0.936 | 99.80 | 22.69 | 0.941 | 81.33 |
| Actor07 | 24.10 | 0.936 | 111.91 | 25.08 | 0.939 | 122.06 | 24.96 | 0.943 | 112.92 | 24.22 | 0.937 | 103.29 |

Table G: **Results on the MVHumanNet sequences.** Our method demonstrates superior novel view synthesis results in modeling loose clothing with over **17.5%** improvement in LPIPS.

| | 102107 | | | 200173 | | | 204129 | | | Avg | | |
|---|---|---|---|---|---|---|---|---|---|---|---|---|
| | PSNR↑ | SSIM↑ | LPIPS↓ | PSNR↑ | SSIM↑ | LPIPS↓ | PSNR↑ | SSIM↑ | LPIPS↓ | PSNR↑ | SSIM↑ | LPIPS↓ |
| HumanNeRF | 23.62 | 0.964 | 48.24 | 22.48 | 0.944 | 71.01 | 23.10 | 0.960 | 47.09 | 23.07 | 0.956 | 55.45 |
| MonoHuman | 23.55 | 0.965 | 56.25 | 21.37 | 0.942 | 79.74 | 23.00 | 0.961 | 50.64 | 22.64 | 0.956 | 62.21 |
| PoseVocab | 21.89 | 0.961 | 52.02 | 19.69 | 0.932 | 94.75 | 20.99 | 0.955 | 61.49 | 20.85 | 0.949 | 69.42 |
| Vid2Avatar | 24.32 | 0.967 | 49.91 | 20.85 | 0.943 | 81.58 | 23.77 | 0.964 | 49.47 | 22.98 | 0.958 | 60.32 |
| 3DGS-Avatar | 25.29 | 0.967 | 47.49 | 22.73 | 0.947 | 77.30 | 23.21 | 0.961 | 48.90 | 23.74 | 0.958 | 57.90 |
| GauHuman | 25.02 | 0.967 | 60.02 | 22.92 | 0.946 | 97.76 | 23.35 | 0.960 | 60.92 | 23.76 | 0.957 | 72.90 |
| **Ours** | 24.80 | 0.970 | 40.01 | 22.15 | 0.945 | 63.32 | 23.80 | 0.964 | 40.28 | 23.56 | 0.960 | 47.18 |

Tab. G respectively. Being consistent with the results in ActorsHQ sequences, our method stably improves baselines in LPIPS metric with over **17.5%** relative improvement.

**Pseudo Ground Truth Shape.** As shown in Fig. O, the pseudo ground truth shape reconstructed as ARAH (Wang et al., 2022a) smooths out the surface details and introduces unwanted artifacts (e.g. the ground surfaces). Thus the imperfect pseudo ground truth shapes benefit Vid2Avatar in most sequences with a slightly higher **Normal Consistency** metrics in Tab. I than ours.

**Training Vid2Avatar with Masks.** The original Vid2Avatar paper asserts that the simultaneous learning of separating humans from any background and reconstructing intricate avatar surfaces is pivotal. However, in ZJU-Mocap sequences, where the background is predominantly black and certain foreground elements appear dark, the task of precisely extracting the foreground and synthesizing texture details becomes notably more challenging. To better compare with this SDF-based volume rendering baseline, we thus impose the ground truth supervision and explicitly extract the foreground by applying the mask for network training. We denote the original version and the mask enhanced version as $V2A^*$ and $V2A_{mask}$ respectively and list their performance on ZJU-MoCap dataset in Tab. K. We can see that the mask enhanced model constantly outperforms the original implementation across all sequences, proving our configuration effectiveness. Fig. P shows extra visual results to comply with the aforementioned findings. Due to the overall superior qualitative and quantitative outputs from the mask enhanced Vid2Avatar, we only report the results of $V2A_{mask}$ in experiments.

**Visualization of non-rigid motions.** In Fig. M, we demonstrate our capability in producing pose-dependent motions by revealing our dynamic wrinkle synthesis across different frames. Specifically, irregular wrinkles on the pant gradually vanish as human moves. Note that, how to completely achieve faithful reproduction of the pose-dependent patterns is still an open problem in the community, which also lies in our future work.

**Summary of Supplementary Video.** In the supplementary video, we provide comparison results on the ZJU-Mocap dataset, the Youtube sequence, the SynWild dataset, the ActorsHQ dataset and MVHumanNet dataset. We render videos under the novel view setting for the MVHumanNet dataset

Table H: **The unseen pose results on the in-the-wild videos.** Our locality sensitive avatar representations enable notably superior perceptual quality than chosen baselines and achieve best scores across almost all metrics.

| | Story | | | Invisible | | | Oleks | | | Weipeng | | | Avg | | |
|---|---|---|---|---|---|---|---|---|---|---|---|---|---|---|---|
| | LPIPS↓ | FID↓ | KID↓ | LPIPS↓ | FID↓ | KID↓ | LPIPS↓ | FID↓ | KID↓ | LPIPS↓ | FID↓ | KID↓ | LPIPS↓ | FID↓ | KID↓ |
| HumanNeRF | 31.35 | 63.28 | 24.94 | 33.72 | 72.29 | 36.12 | 28.43 | 63.71 | 25.49 | 20.23 | 37.94 | 2.59 | 28.83 | 59.81 | 22.29 |
| MonoHuman | 32.73 | 65.23 | 27.20 | 34.39 | 79.94 | 40.30 | 27.56 | 67.17 | 31.48 | 21.48 | 44.18 | 4.93 | 29.04 | 64.13 | 25.98 |
| NPC | 29.59 | 53.62 | 13.72 | 35.28 | 80.17 | 43.99 | 31.41 | 70.26 | 30.08 | 23.36 | 46.50 | 241.17 | 29.91 | 62.64 | 82.24 |
| Vid2Avatar | 36.85 | 187.24 | 250.34 | 40.52 | 198.51 | 276.45 | 31.36 | 165.90 | 178.27 | 23.66 | 205.27 | 12.67 | 33.10 | 189.23 | 179.43 |
| PM-Avatar | 35.15 | 78.86 | 45.86 | 42.67 | 109.49 | 82.58 | 34.44 | 78.38 | 47.96 | 29.50 | 70.17 | 42.44 | 35.44 | 84.22 | 54.71 |
| **Ours** | 27.99 | 56.89 | 16.70 | 30.98 | 69.65 | 36.08 | 26.08 | 55.71 | 18.87 | 19.24 | 40.02 | 3.28 | 26.07 | 55.57 | 18.73 |

Table I: **Geometry reconstruction comparisons on ZJU-Mocap (Peng et al., 2021b).** Following ARAH (Wang et al., 2022a), we compute the L2 Chamfer Distance (CD) and Normal Consistency (NC) for geometry evaluations. Our method demonstrates superior capability in synthesizing high-quality shapes with fine-grained details. Note that the imperfect pseudo-ground-truth meshes smooth out many necessary geometry and hinder us from achieving greater quantitative improvements respecting the NC metrics as examplified in Fig. O. Scores are colored as in Tab. A.

| | HumanNeRF | | MonoHuman | | NPC | | Vid2Avatar | | PM-Avatar | | 3DGS-Avatar | | GoMAvatar | | **Ours** | |
|---|---|---|---|---|---|---|---|---|---|---|---|---|---|---|---|---|
| | CD ↓ | NC ↑ | CD ↓ | NC ↑ | CD ↓ | NC ↑ | CD ↓ | NC ↑ | CD ↓ | NC ↑ | CD ↓ | NC ↑ | CD ↓ | NC ↑ | CD ↓ | NC ↑ |
| S313 | 0.025 | 0.820 | 0.119 | 0.685 | 0.040 | 0.821 | 0.020 | 0.920 | 0.040 | 0.815 | 0.074 | 0.738 | 0.016 | 0.887 | 0.022 | 0.907 |
| S377 | 0.041 | 0.778 | 0.036 | 0.782 | 0.050 | 0.802 | 0.034 | 0.863 | 0.032 | 0.821 | 0.053 | 0.729 | 0.036 | 0.828 | 0.032 | 0.859 |
| S386 | 0.037 | 0.779 | 0.060 | 0.716 | 0.070 | 0.748 | 0.033 | 0.881 | 0.056 | 0.785 | 0.093 | 0.679 | 0.028 | 0.852 | 0.028 | 0.866 |
| S387 | 0.053 | 0.750 | 0.047 | 0.732 | 0.048 | 0.766 | 0.022 | 0.876 | 0.041 | 0.748 | 0.076 | 0.699 | 0.029 | 0.832 | 0.023 | 0.864 |
| S390 | 0.032 | 0.769 | 0.054 | 0.745 | 0.050 | 0.752 | 0.024 | 0.848 | 0.041 | 0.760 | 0.068 | 0.689 | 0.033 | 0.809 | 0.025 | 0.847 |
| S392 | 0.084 | 0.715 | 0.076 | 0.720 | 0.083 | 0.720 | 0.082 | 0.755 | 0.066 | 0.711 | 0.085 | 0.644 | 0.077 | 0.746 | 0.079 | 0.754 |
| S393 | 0.071 | 0.745 | 0.065 | 0.748 | 0.072 | 0.722 | 0.061 | 0.823 | 0.069 | 0.718 | 0.099 | 0.672 | 0.067 | 0.785 | 0.063 | 0.817 |
| S394 | 0.065 | 0.769 | 0.062 | 0.769 | 0.079 | 0.765 | 0.056 | 0.853 | 0.063 | 0.768 | 0.085 | 0.709 | 0.064 | 0.823 | 0.057 | 0.844 |
| Avg | 0.051 | 0.765 | 0.065 | 0.737 | 0.061 | 0.762 | 0.042 | 0.852 | 0.051 | 0.766 | 0.079 | 0.695 | 0.044 | 0.820 | 0.041 | 0.845 |

Table J: **Geometry reconstruction comparisons on SynWild (Guo et al., 2023).** Following Vid2Avatar, we use the semi-synthetic SynWild dataset for geometry evaluations. Our method produces overall best results over baselines. Note that the imperfect human pose estimation prevents us from achieving fully accurate quantitative evaluations compared to Vid2Avatar. Although we try our best to pick representative images in front and back views for MonoHuman[†], the model still cannot converge for the first sequence. Thus we ignore Monohuman when measuring mean values of Chamfer distance (CD) and normal consistency (NC). Scores are colored as in Tab. A.

| | HumanNeRF | | MonoHuman[†] | | NPC | | Vid2Avatar | | PM-Avatar | | 3DGS-Avatar | | GoMAvatar | | **Ours** | |
|---|---|---|---|---|---|---|---|---|---|---|---|---|---|---|---|---|
| | CD ↓ | NC ↑ | CD ↓ | NC ↑ | CD ↓ | NC ↑ | CD ↓ | NC ↑ | CD ↓ | NC ↑ | CD ↓ | NC ↑ | CD ↓ | NC ↑ | CD ↓ | NC ↑ |
| S1 | 0.644 | 0.620 | N/A | N/A | 0.602 | 0.614 | 0.609 | 0.662 | 0.607 | 0.621 | 0.637 | 0.563 | 0.957 | 0.605 | 0.623 | 0.658 |
| S2 | 0.695 | 0.613 | 0.714 | 0.595 | 0.735 | 0.604 | 0.745 | 0.668 | 0.751 | 0.620 | 0.838 | 0.553 | 0.739 | 0.653 | 0.745 | 0.666 |
| S3 | 0.569 | 0.638 | 0.527 | 0.639 | 0.584 | 0.619 | 0.534 | 0.693 | 0.553 | 0.653 | 0.610 | 0.559 | 0.543 | 0.674 | 0.521 | 0.695 |
| S4 | 0.324 | 0.635 | 0.358 | 0.608 | 0.306 | 0.606 | 0.316 | 0.694 | 0.316 | 0.615 | 0.331 | 0.564 | 0.318 | 0.675 | 0.283 | 0.695 |
| S5 | 0.302 | 0.673 | 0.301 | 0.609 | 0.288 | 0.633 | 0.293 | 0.718 | 0.274 | 0.654 | 0.351 | 0.560 | 0.277 | 0.700 | 0.254 | 0.733 |
| Avg | 0.507 | 0.635 | 0.475 | 0.612 | 0.503 | 0.615 | 0.499 | 0.687 | 0.500 | 0.632 | 0.553 | 0.560 | 0.567 | 0.661 | 0.485 | 0.690 |

while all other results are driven by novel poses. Except Youtube sequences, we utilize the human poses provided by official datasets as model inputs. Instead, we use poses from AIST++ (Li et al., 2021a) to animate our avatar representations in the YouTube setting, showcasing our robustness to in-the-wild videos. We further generate normal map comparisons with Vid2Avatar to highlight our capability in synthesizing fine-grained geometric details for SynWild images. Outputs from the rigid deformation and full model are also illustrated to reveal the effects of non-rigid offsets.

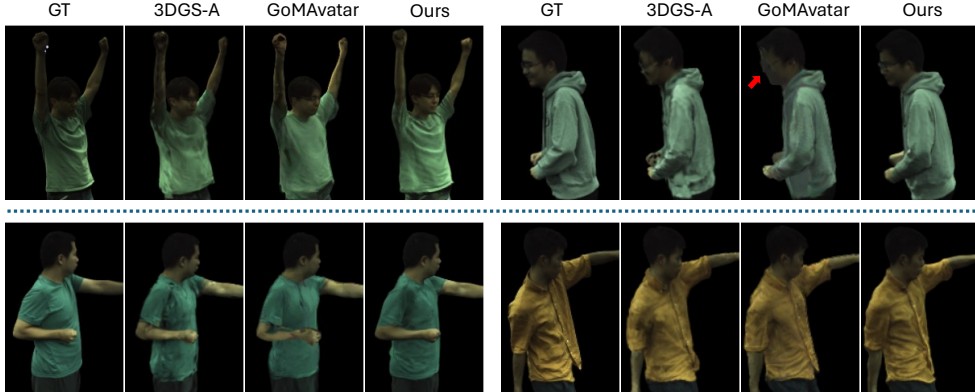

Figure D: **Visual comparisons on ZJU-Mocap** with Gaussian Splatting framework. Compared to the chosen baselines, our method can better preserve the sharp wrinkles (top row) without artifacts. In contrast, the baselines either blur the desired cloth wrinkles and highlighted facial structures, and yield unwanted black distortion patterns across almost all examples. Here we use "3DGS-A" to represent the "3DGS-Avatar" baseline for short. The results for novel view synthesis and novel pose rendering are shown in both rows respectively.

Table K: **Evaluating importance of training Vid2Avatar with mask.** Extracting masks for training, Vid2Avatar produces better quantitative metrics across almost all subjects in all three metrics. V2A* and $V2A_{mask}$ denote the original Vid2Avatar model and mask enhanced model respectively.

| | S313 | | | S377 | | | S386 | | | S387 | | | S390 | | | S392 | | | S393 | | | S394 | | |
|---|---|---|---|---|---|---|---|---|---|---|---|---|---|---|---|---|---|---|---|---|---|---|---|---|
| | PSNR↑ | SSIM↑ | LPIPS↓ | PSNR↑ | SSIM↑ | LPIPS↓ | PSNR↑ | SSIM↑ | LPIPS↓ | PSNR↑ | SSIM↑ | LPIPS↓ | PSNR↑ | SSIM↑ | LPIPS↓ | PSNR↑ | SSIM↑ | LPIPS↓ | PSNR↑ | SSIM↑ | LPIPS↓ | PSNR↑ | SSIM↑ | LPIPS↓ |
| **Novel View Synthesis** | | | | | | | | | | | | | | | | | | | | | | | | |
| V2A* | 26.82 | 0.913 | 98.74 | 29.51 | 0.951 | 70.62 | 31.16 | 0.944 | 76.46 | 28.09 | 0.938 | 83.55 | 28.07 | 0.923 | 92.48 | 30.37 | 0.943 | 82.93 | 27.19 | 0.929 | 83.42 | 28.95 | 0.931 | 81.31 |
| $V2A_{mask}$ | **28.44** | **0.966** | **37.74** | **29.52** | **0.975** | **24.91** | **33.04** | **0.977** | **30.65** | **28.15** | **0.965** | **40.26** | **30.73** | **0.970** | **37.33** | **30.46** | **0.971** | **36.58** | **27.94** | **0.962** | **40.93** | **29.82** | **0.965** | **36.46** |
| **Novel Pose Rendering** | | | | | | | | | | | | | | | | | | | | | | | | |
| V2A* | 26.67 | 0.913 | 97.79 | **29.86** | 0.953 | 68.45 | 30.85 | 0.941 | 77.00 | **27.42** | 0.938 | 80.78 | 28.88 | 0.926 | 88.08 | 30.20 | 0.942 | 84.57 | 27.20 | 0.929 | 82.50 | 28.54 | 0.932 | 78.80 |
| $V2A_{mask}$ | **28.00** | **0.965** | **38.43** | 29.80 | **0.977** | **24.77** | **32.48** | **0.975** | **32.43** | 27.34 | **0.964** | **40.83** | **30.81** | **0.971** | **34.73** | **30.50** | **0.971** | **37.53** | **28.09** | **0.962** | **40.17** | **29.24** | **0.963** | **36.61** |

# E  DISCUSSIONS & FUTURE WORKS

Although our method can achieve state-of-the-art results in novel image rendering and 3D avatar reconstruction, the dense MLP computation within the volume rendering operation limits real-time applications, posing our most significant constraint. To address this issue, some works implement grid-based methods (Müller et al., 2022; Chen et al., 2022; Wu et al., 2023) and adapt the modern Gaussian Splatting framework (Kerbl et al., 2023a; 2024) to improve both training and inference efficiency. However, note that there is always a speed-accuracy tradeoff and our emphasis is on detail reproduction and model generalization. How to extend our locality sensitive avatar representations to different image synthesis pipelines (e.g. 3D Gaussian Splatting) would be our next research direction in the future.

Since our method is designed upon NeRF, it requires individual training for each actor and cannot generalize to other humans without additional training. Developing an algorithm to infer multiple actors without training from scratch comes to be an intriguing direction. Enhancing our framework with pattern editing features, which we do not explicitly consider currently, is also our future work.

As shown in Fig. Q, our method, like all others, struggles under extreme conditions where test poses differ significantly from training poses. This results in distorted body contours and inaccurate texture reproduction. Addressing these generalization issues for challenging avatar poses remains an open problem. For example, future research in areas such as domain adaptation and generative models may provide valuable insights and solutions to enhance the performance and generalization capabilities of neural avatar models in extreme conditions.

Another promising direction for future work is extending our framework to better model loose clothing. As shown in Fig. R, our method generally produces visually appealing results, whereas baselines often twist the character's body and introduce floating artifacts. However, our approach

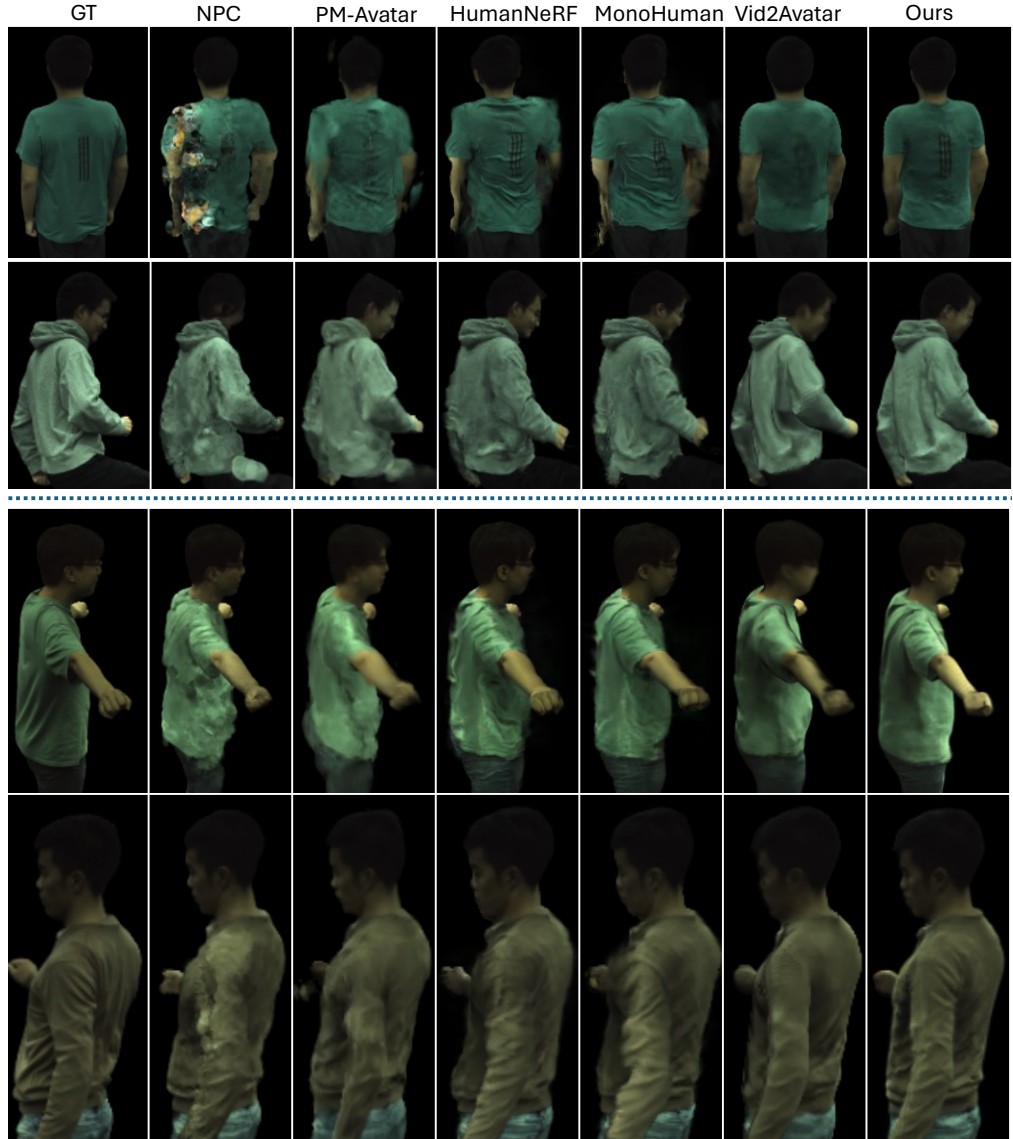

| GT | NPC | PM-Avatar | HumanNeRF | MonoHuman | Vid2Avatar | Ours |

Figure E: **Visual comparisons on ZJU-Mocap.** Our method can maintain large-scale body shapes and finely matched textures simultaneously while baselines either blur the notable patterns or greatly twist shape outlines under novel camera views (upper row) and novel avatar poses (bottom row).

struggles to accurately reconstruct human faces due to hair movement. Consequently, modeling the motion of soft objects and loose clothing will be a key focus of our future research.

## F  SOCIAL IMPACTS

Our research has the potential to significantly enhance the efficiency of human avatar modeling pipelines, alleviating the issues of underrepresented individuals and activities in supervised datasets. However, the ease of use, requiring only a single video as input, also poses the risk of generating 3D models without proper consent and ethical considerations. Therefore, it is crucial to adhere to strict consent and ethical guidelines before utilizing our algorithm.

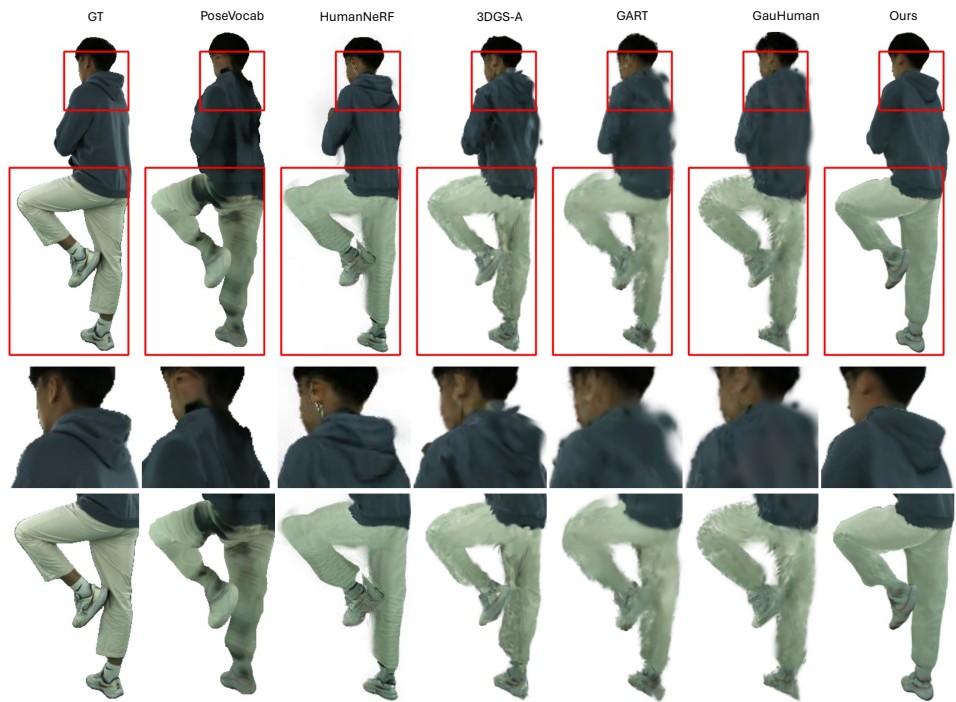

Figure F: **Comparisons on MVHumanNet (Part 1).** Our method presents superior shape outlines (e.g. highlighted legs and hats) while baselines notably blur the boundaries with noisy artifacts.

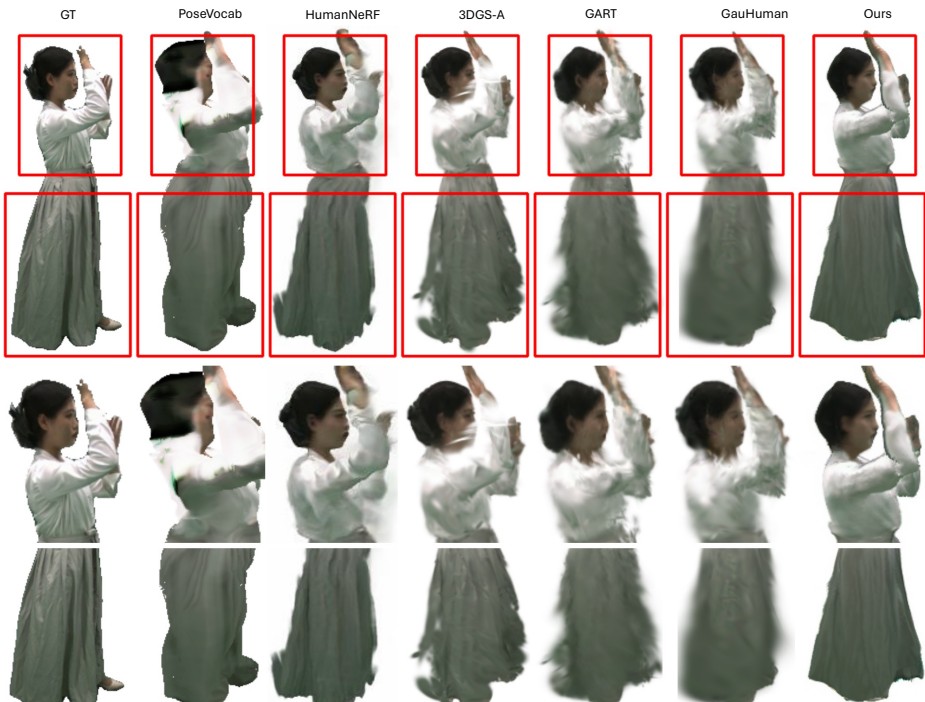

Figure G: **Comparisons on MVHumanNet (Part 2).** Our method can generalize better to loose skirts and challenging skeleton poses. In comparison, baselines either cannot preserve reasonable outlines, or seriously distort surfaces into discontinuous parts with spiky artifacts near boundary.

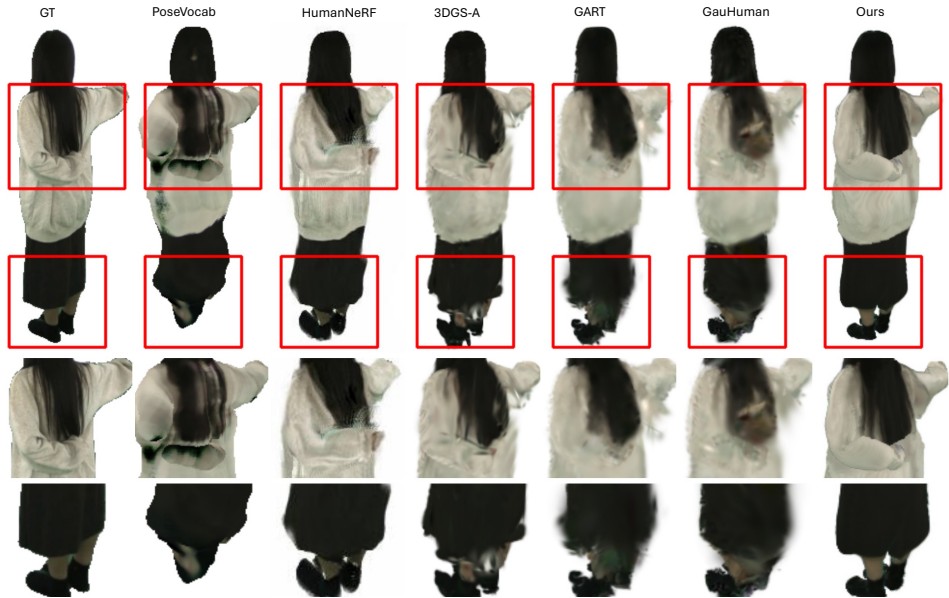

Figure H: **Rendering Comparisons on MVHumanNet (Part 3).** While the baselines blend the left hand with the loose cloth and merge the two feet into a single structure, our method effectively separates them and more accurately generates fine details such as tiny hairs.

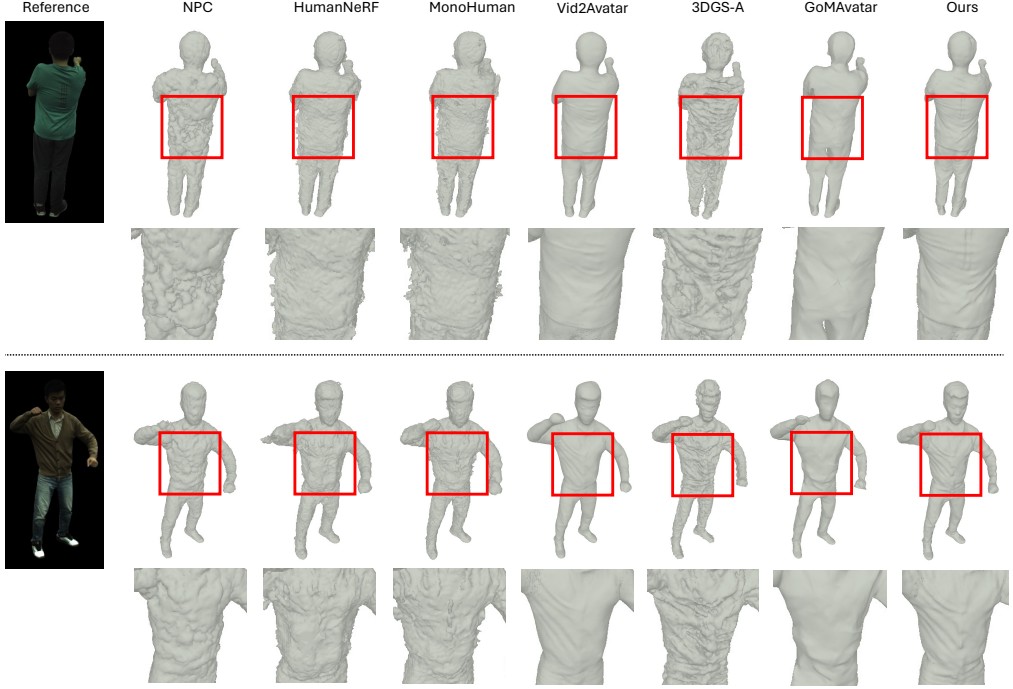

Figure I: **Geometry reconstruction on ZJU-Mocap.** Our method can generate better geometric structures and adaptive fine-grained details than all baselines. In comparison, the baselines either significantly distort geometric patterns with noise or weaken notable cloth wrinkles. Here we refer to "3DGS-Avatar" as "3DGS-A" for short.

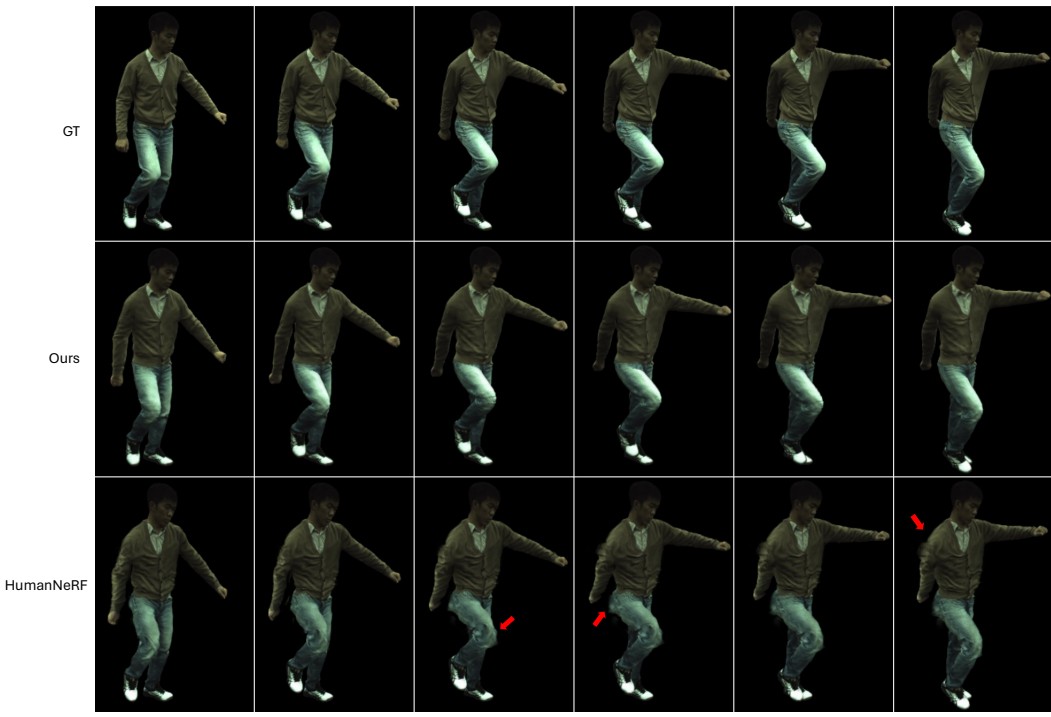

Figure J: **Comparisons on Multiple Video Frames.** Our method preserves realistic rendering outputs across consecutive video frames. In contrast, HumanNeRF introduces unwanted blurry artifacts as highlighted by red arrows occasionally.

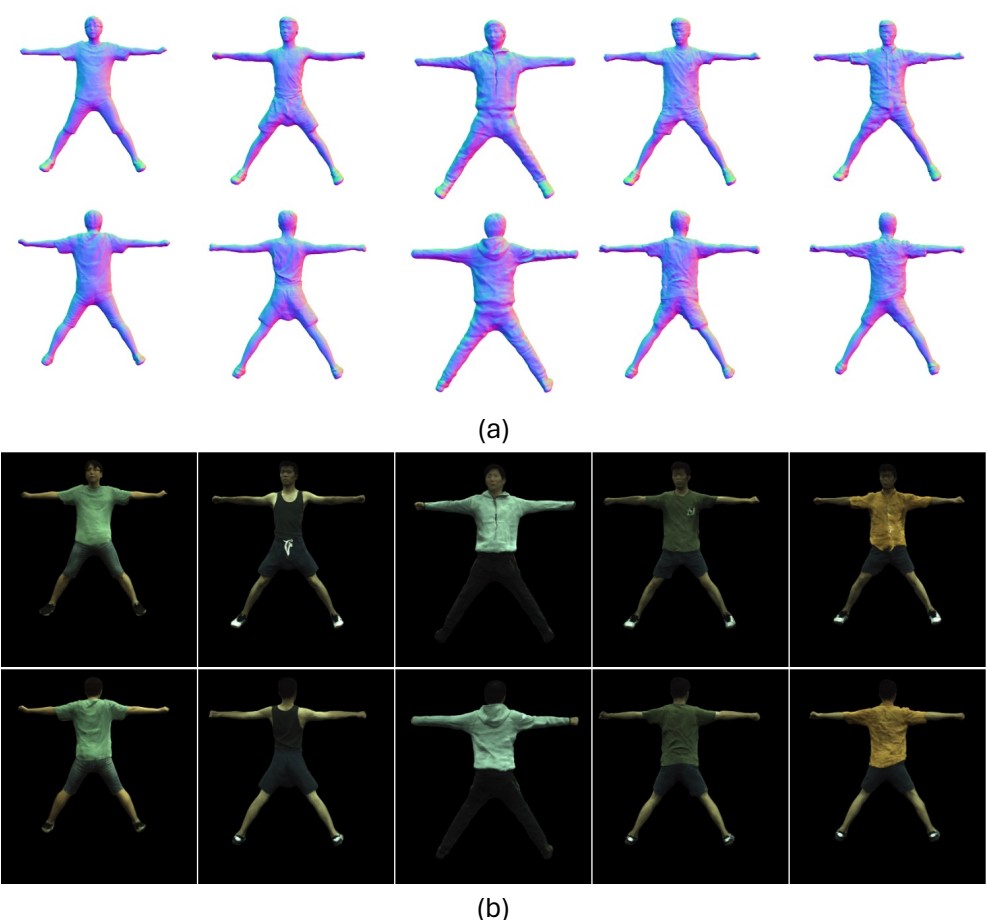

Figure K: Canonical normal maps (a) and appearances (b) for ZJU-MoCap sequences.

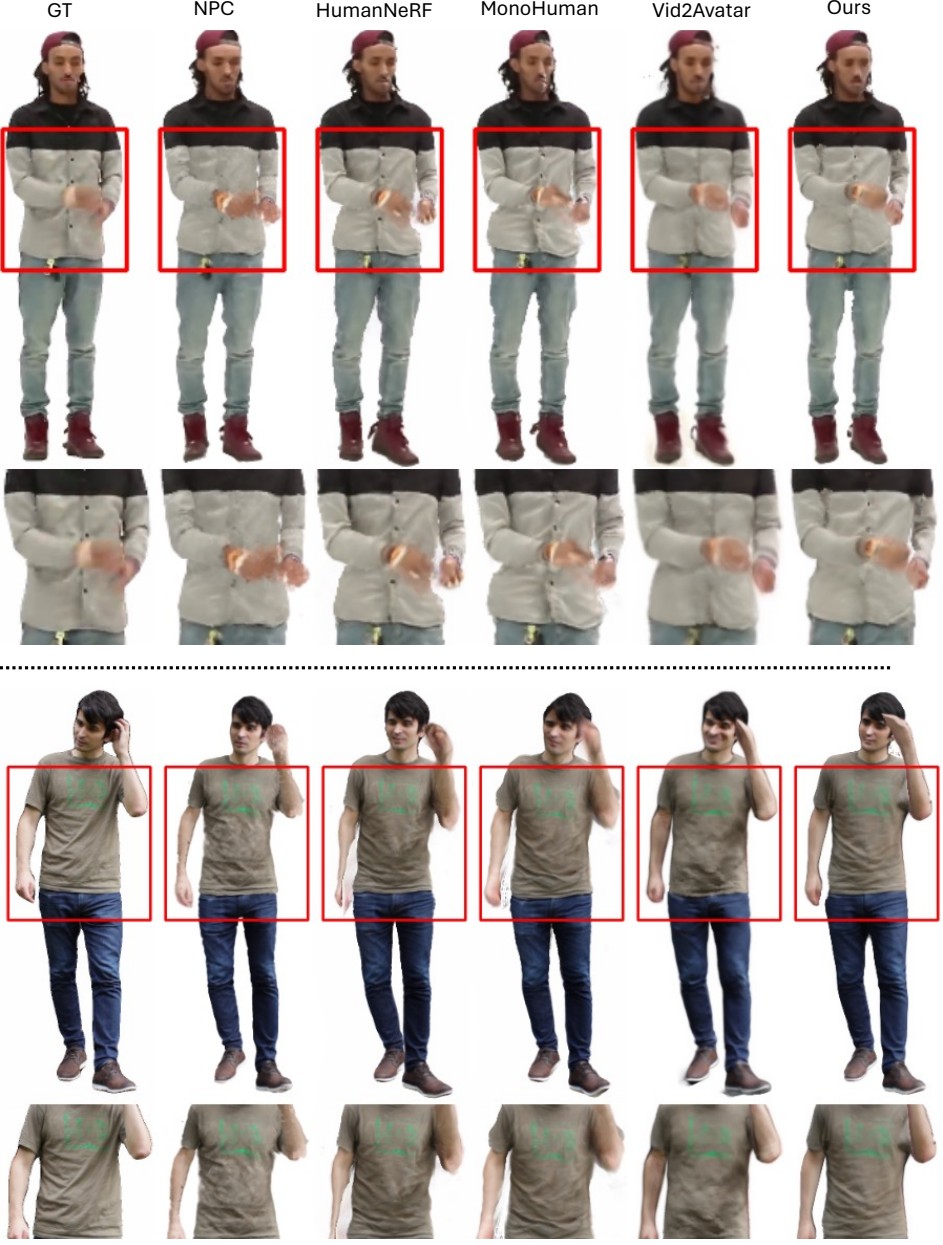

Figure L: **Novel pose rendering on Youtube sequences.** Our method can best preserve shape contours and produce most realistic cloth appearances (e.g. the wrinkles shown in $2^{nd}$ row) while baselines distort either produced details (e.g. NPC) or body shapes (e.g. HumanNeRF).

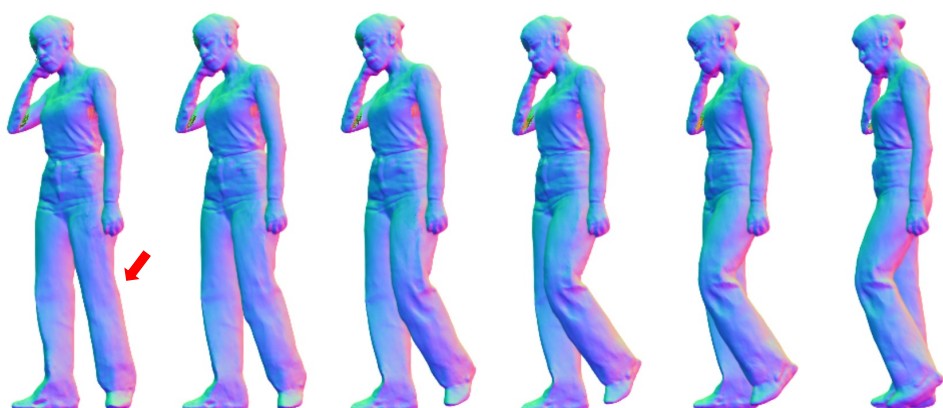

Figure M: **Visualization of non-rigid effects.** As highlighted by the red arrow, the wrinkles near the knee increase when the subject bends her legs.

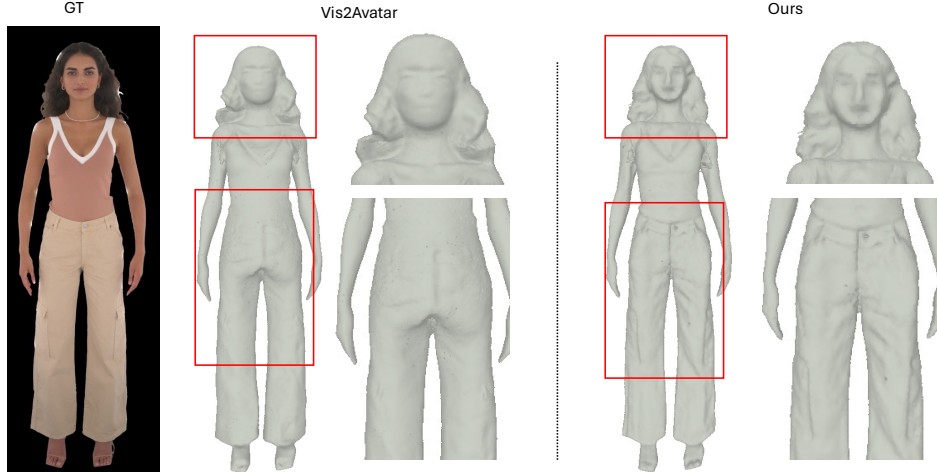

Figure N: **Geometry reconstruction on ActorsHQ.** Our method improves existing baselines by preserving more perceptible facial patterns, more clear pant boundary and delicate pant wrinkles.

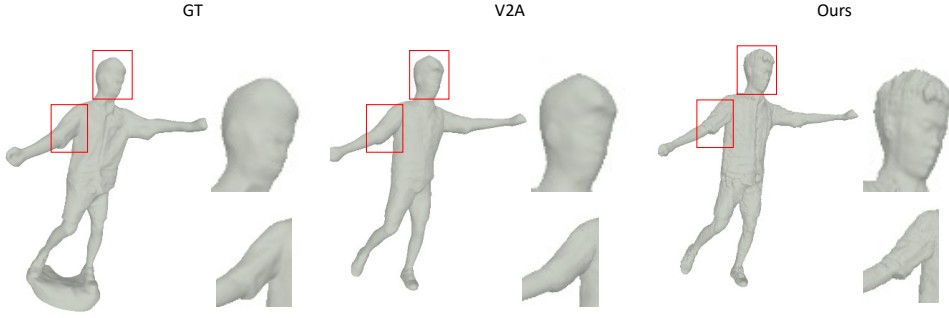

Figure O: **Visualization for pseudo ground truth shape.** Both shapes from pseudo ground truth (GT) and Vid2Avatar (V2A) smooth over the geometric details (e.g. the highlighted facial expressions and shoulder). Moreover, the pseudo GT shape also introduces the unwanted ground surfaces on the bottom, benefiting Vid2Avatar more from computing quantitative metrics.

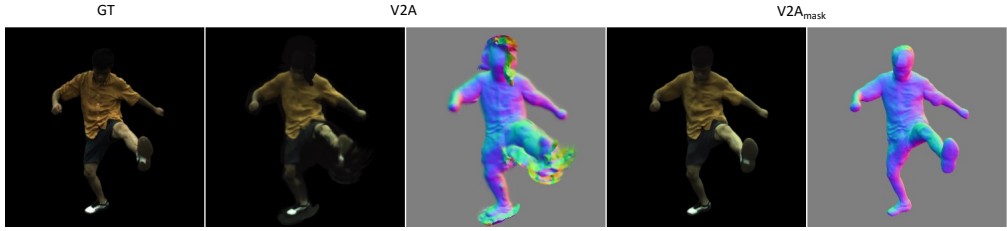

Figure P: **Training Vid2Avatar with Masks.** The extracted masks significantly improve the original Vid2Avatar's capability in preserving stable contours and avoiding artifacts.

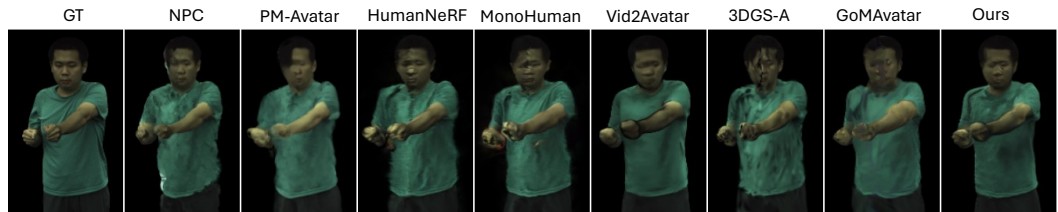

Figure Q: **Failure cases on out-of-distribution poses.** Generalizing to challenging cases remains an open problem, as all methods struggle to maintain visually pleasing textures and accurate body outlines under such poses.

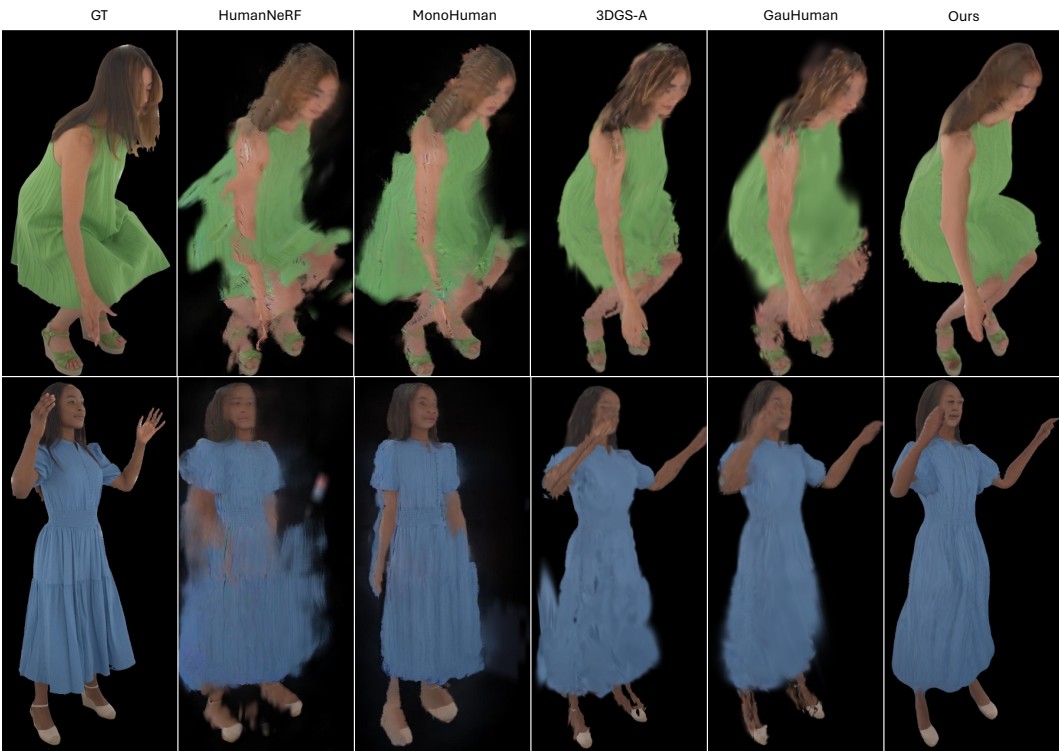

Figure R: **Failure cases on loose-fitting cloth.** While our method produces more satisfactory images with consistent body contours and realistic fine details compared to state-of-the-art baselines, accurately modeling loose clothing in monocular video settings remains a challenge for future work.

