# OpenReview forum: "Locality Sensitive Avatars From Video"
_ICLR.cc/2025/Conference — ICLR 2025 Poster_

### Official Review · Reviewer_cJiu · 2024-10-30

**Soundness:** 3
**Presentation:** 2
**Contribution:** 2
**Rating:** 6
**Confidence:** 3

**Summary:**

This paper studies the problem of learning 3D neural avatars from 2D videos. It is based on several prior works that decompose the dynamic neural human avatar into canonical space, rigid deformation, and non-rigid deformation. The major contribution lies in the modeling of the non-rigid deformation part. Specifically, it trains a pose-conditioned non-rigid deformation prediction network with a GNN architecture. Some experiments show improvement of the proposed method.

**Strengths:**

- The studied problem in this paper is important. Reconstructing neural human avatars from videos has been a long-standing problem in the community, with several different applications in the movie industry, AR/VR, and gaming industry.
- The proposed method basically follows the framework set by the prior works, which is sound. Modeling the non-rigid deformation with a pose-conditioned network is also a sound and widely used method in the community.

**Weaknesses:**

- The major concern is the generalizability of the pose-conditioned GNN. The multi-view human data is sparse in the community, therefore the GNN might overfit on those training and struggle to generalize to the in-the-wild dataset.
- Most results presented are with tight clothes, which might not fully demonstrate the effectiveness of the proposed method. It would be interesting to see more large-scale results on more challenging clothes, such as THUman2.1 dataset.
- The major contribution of this paper can be seen as incremental. The only major change of this paper's pipeline compared to prior works is to use GNN to model non-rigid deformation, which is a sound but not-so-exciting technical contribution.
- Some baselines are not compared, e.g. [Li et al. 2023], so it's hard to say whether the proposed method is truly the state-of-the-art.
- Some experiment settings are not clear. See questions for details.

[Li et al. 2023] PoseVocab: Learning Joint-structured Pose Embeddings for Human Avatar Modeling.

**Questions:**

- For the experiments on the ActorsHQ dataset, is it rendered under a novel pose or training pose?
- For the videos rendered in the supplementary material, is it under novel pose or training pose?

---

> ### Author Response · Authors · 2024-11-22
> **Response to Reviewer cJiu**
>
> We thank the reviewer for reviewing our paper and providing valuable comments. We address the concerns in the common responses and below, and are open to further discussion and questions.
>
> **Q1: Concerning the generalizability of the pose-conditioned GNN**
>
> Thank you for your insightful comments. However, please note that this point is well aligned with our key motivation. As summarized in Table 1 and discussed in the **Introduction** Section (L47-L52), some state-of-the-art methods (e.g. NPC and PM-Avatar) encode locality information with GNN to enhance detail synthesis but struggle to generalize well and thus yield artifacts with sparse input data. In contrast, other methods propose to learn the geometry and appearance in canonical space to preserve more consistent outputs across different frames but are inferior in producing fine-grained signals. Figure B provides an example that unifying these two ideas can best adopt their strengths and eliminate their weaknesses, while the GNN-based baseline (NPC) generates significant unwanted black artifacts. The results of PoseVocab shown below also evidence this shortcoming.
>
> Our ablation study results further qualitatively and quantitatively verify this finding in Figure 8 and Table 3 (b). After removing the canonical feature learning, the "w / o canonical" baseline fails to maintain reasonable body contours and significantly distorts the vertical strip textures. In contrast, our full model equipped with locality encoding and canonical space rendering can faithfully reconstruct the ground truth image. In Figure 5, Table F and the supplementary video, our empirical improvements in in-the-wild sequences additionally display our generalization to real-world scenarios.
>
> **Q2: Results on more challenging clothes, such as THUman2.1 dataset**
>
> Thank you for your suggestions. As presented in the **Common Response to Reviewers**, we have shown additional results on one ActorsHQ sequence and three MVHumanNet sequences. Together with results on three ActorsHQ sequences shown in the original manuscript, we have utilized at least six sequences to highlight our ability to handle loose clothing. We have also contacted the authors of the THUman2.1 dataset for download access but found that it focuses on high-quality human scans rather than images captured by multiple cameras which do not fit our task.
>
> **Q3: The major contribution of this paper can be seen as incremental**
>
> As summarized in Table 1 of our manuscript, our method makes the first attempt to incorporate both locality encoding and canonical space rendering into one single framework while previous methods only model one of these two objectives with inferior output quality. We emphasize that our focus is to verify the effectiveness of our observation rather than designing fancy components. One key advantage of following the existing methods, including NPC and PM-Avatar, to apply GNN for locality encoding is that we can enable a simple and neat network and rule out the effects of irrelevant factors such that our key motivation can be clearly supported.
>
> Empirically, the significance of our method can be clearly shown via the relative improvements upon the previous methods and our extensive ablation studies, highlighting our technical contribution.
>
> **Q4: Comparisons with PoseVocab**
>
> Thank you for your suggestion. As recommended, we have evaluated PoseVocab with its default settings on the MVHumanNet dataset. The quantitative metrics are provided below and our model consistently outperforms PoseVocab across all metrics.
>
> | | PoseVocab | Ours |
> |:-----:|:---------:|:-----:|
> | PSNR | 20.85 | **23.56** |
> | SSIM | 0.949 | **0.959** |
> | LPIPS | 69.42 | **49.44** |
>
> Consistent with your comments on the generalizability of pose-conditioned GNNs, PoseVocab tends to overfit and struggles to generalize effectively to monocular videos. As a result, it significantly distorts continuous shape contours and warps the desired foreground patterns, as shown in Figure F, Figure G and Figure H. In contrast, our method can successfully reproduce signals in multiple scales, demonstrating the effectiveness of integrating locality encoding with canonical space rendering.
>
> **Q5: Experiment setting on the ActorsHQ dataset**
>
> As described in the “ActorHQ Dataset” part of Section B, we use the images captured by ‘camera 127’ and ‘camera 128’ as training and evaluation data respectively as they can cover the whole human body. Specifically, we utilize the first 75\% images for training and the remaining 25\% images as novel pose references. The overall and per-sequence metrics for both novel view synthesis and novel pose animation are listed in Tab. 3 (b) and Tab. E respectively. The video results in the updated supplementary material are rendered for novel pose animation.

---

> > ### Comment · Reviewer_cJiu · 2024-12-03
> >
> > The reviewer thanks the authors for their rebuttal. I have carefully re-read the original paper and the rebuttal and appreciate the authors' clarifications and efforts in addressing concerns.
> >
> > First of all, the major concerns in the original review have been mostly addressed in this rebuttal, therefore the reviewer would raise the score to 6. However several minor concerns remain, which prevents the reviewer from giving a higher score:
> >
> > **Q1. concerns on the generalizability:**
> >
> > The reviewer agree that the proposed method demonstrated better generalizability in novel poses compared to prior approaches. However the major concern here is the problem setting itself: covering enough pose in the training video is practically impossible, and there might be a lot of out-of-distribution poses if we really want to use the generated model to perform animation tasks. Yes, the proposed method could get acceptable renderings in poses similar to the training poses, as in the problem setting considered in this paper, especially in the sequences of a controlled dataset. However, if we perform motion transfer (say, let the avatar dance), whether the animation is still realistic is questionable.
> >
> > **Q2. Results on more challenging scenarios:**
> >
> > The reviewer agree that the proposed method surpasses the baselines in complex scenarios involving loose clothes. However there is still notable artifacts, such as issues that can be seen in Figure R. That being said, the reviewer agree that this problem is hard to tackle and would like to see follow-up works on this.
> >
> > **Q3: The major contribution:**
> >
> > Thanks for the clarification! Could you elaborate more on why NPC does not have the properties described in this paper? According to my understanding it also has canonical space modeling and a GNN to predict local deformation. What specifically enables this method to outperform NPC?
> >
> > **Additional Comments on Exposition**
> >
> > Moreover, upon another reading,  I strongly feel that the writing and presentation need significant improvement. The introduction relies heavily on technical jargon and assumes too much prior knowledge, making it inaccessible to a broader ICLR audience. The overall flow of the paper is incoherent, which makes it challenging to follow. The technical figures lack polish and could be significantly improved for clarity.
> >
> > While I will not actively champion the acceptance of this paper, I also do not oppose it. This work shows great potential, and with a major revision to the writing and the incorporation of discussions from the rebuttal, it could become significantly stronger.

---

> > > ### Author Response · Authors · 2024-12-03
> > > **Thank you for your response**
> > >
> > > Dear Reviewer cJiu,
> > >
> > > Thank you very much for your detailed comments and raising your score, which is very helpful!
> > >
> > > Generally, while we outperform existing baselines under the setting of novel pose animation and loose cloth modeling, we agree with the reviewer that how to fully address these issues is still an open problem, which drives our future work.
> > >
> > > In terms of comparisons with NPC, our empirical improvement mainly comes from the canonical space rendering. Yes, NPC also utilizes canonical parameters and GNN to enhance detail synthesis. But like other 3DGS based methods, NPC maps the canonical points to per-frame observation space which makes it more vulnerable to sparse data. In comparison, we follow the idea of rendering in canonical space rather than in per-frame observation space to significantly facilitate generalization to monocular videos. While former methods based on canonical space rendering, like HumanNeRF, are prone to distort structured patterns, we simply introduce the local deformation to improve detail synthesis. We would emphasize that, while both canonical space rendering and local deformation are investigated by previous papers, we firstly unify them into one single framework to best adopt their strengths and eliminate their weaknesses which is a significant overlook by the community. These things are summarized in Table 1 of our manuscript. As mentioned by the reviewer, our simple and effective upgrade yields notable experimental improvements across comprehensive evaluation scenarios, revealing our great potentials to future work.
> > >
> > > Also thank you for your comments on our exposition. We will incorporate these suggestions into our future revision.

---

### Official Review · Reviewer_nejY · 2024-11-02

**Soundness:** 3
**Presentation:** 3
**Contribution:** 3
**Rating:** 6
**Confidence:** 4

**Summary:**

This paper introduces Locality-Sensitive Avatar, a novel neural radiance field (NeRF) based network for learning human motions from monocular videos. The key innovation lies in using a graph neural network (GNN) to model non-rigid deformations while preserving fine-grained, locality-sensitive details.

**Strengths:**

The model demonstrates superior rendering performance compared to other approaches, particularly evident through enhanced normal rendering results. It is also capable of learning from a small amount of data and remains effective even for novel poses, showcasing its robustness and versatility.

**Weaknesses:**

The model struggles with accurate reconstruction of detailed areas such as the face or hands. Additionally, the visualization of non-rigid motion is insufficient, making it unclear what this component represents, and it is difficult to clearly understand the difference between considering and not considering non-rigid motion.

The core idea is to utilize non-rigid motion, which could effectively express the fluttering of loose-fit clothes. The current examples capture only person-specific details.

**Questions:**

1. What are the differences between the proposed model and GauHuman or HUGS?
@inproceedings{hu2024gauhuman,
  title={Gauhuman: Articulated gaussian splatting from monocular human videos},
  author={Hu, Shoukang and Hu, Tao and Liu, Ziwei},
  booktitle={Proceedings of the IEEE/CVF Conference on Computer Vision and Pattern Recognition},
  pages={20418--20431},
  year={2024}
}

@inproceedings{kocabas2024hugs,
  title={Hugs: Human gaussian splats},
  author={Kocabas, Muhammed and Chang, Jen-Hao Rick and Gabriel, James and Tuzel, Oncel and Ranjan, Anurag},
  booktitle={Proceedings of the IEEE/CVF conference on computer vision and pattern recognition},
  pages={505--515},
  year={2024}
}

2. Does using SMPL pose parameters as LBS weights result in any artifacts?

3. What is the rendering speed of the proposed model?

4. Can you show the qualitative and quantitative results if the non-rigid Δx component is omitted?

5. Does considering non-rigid deformations help in accurately representing effects like the flapping of clothes?

---

> ### Author Response · Authors · 2024-11-22
> **Response to Reviewer nejY**
>
> We thank the reviewer for reviewing our paper and providing valuable comments. We address the concerns in the common responses and below, and are open to further discussion and questions.
>
> **Q1: The model struggles with accurate reconstruction of detailed areas such as the face or hands**
>
> Thank you for pointing this out. Our method uses SMPL parameters to drive skeletal deformation and represent the skeleton utilized in the GNN, which does not account for fine-grained details of the hands and face. We note that this limitation has been largely overlooked by most methods until the recent publication of EVA [Hu et al. 2024] which leverages the more advanced SMPL-X model for the modeling of hand and face. We would like to emphasize that our proposed approach is not limited to the SMPL model, thus how to more faithfully capture the expressive details of hand and face (e.g. by SMPL-X models) in our framework lies in our future work.
>
> [Hu et al. 2024] Expressive Gaussian Human Avatars from Monocular RGB Video, NeurIPS 2024
>
> **Q2: The visualization of non-rigid motion is insufficient**
>
> Thank you for your comments. The effects of non-rigid motion are widely researched in recent neural avatar modeling literature. For example, in Section 3.1 of Vid2Avatar (Guo et al., 2023), the authors interpret the non-rigid deformations as dynamically changing wrinkles on clothes. To better demonstrate the effect of non-rigid motion, we visualize the normal maps across different frames in Figure M. It is shown that our method successfully animates the wrinkles near the knee when the subject bends her legs.
>
> To further demonstrate the influence of non-rigid motion in our framework, we remove the offset and denote the ablated model with only skeletal deformation as "w / o locality". We present the visual images and quantitative scores in Figure 8 and Table 3 (b) respectively. As illustrated in Figure 8, the vertical stripe texture is considerably blurred by the "w / o locality" model because that texture varies across different frames and only rigid motion does not suffice to faithfully capture these variations. With non-rigid deformations, we can better reproduce the pose-dependent effects without introducing artifacts. However, please note that, how to fully accurately capture the complicated non-rigid deformation of clothes under the monocular setting is still an open problem for all existing methods, which calls for future research.
>
> **Q3: The differences between the proposed model and GauHuman or HUGS?**
>
> There are three key differences between our method and GauHuman or HUGS:
> (1) Similar to other 3DGS based methods summarized in Table 1, GauHuman and HUGS render images in observation space instead of in canonical space, thus being vulnerable to sparse data (e.g. monocular setting).
> (2) Both GauHuman and HUGS directly perform LBS to drive canonical Gaussian representations to given poses without explicitly modeling the non-rigid motion component. In comparison, our method computes the rigid skeletal deformation and corresponding non-rigid deformation simultaneously.
> (3) These two methods are implemented with the 3D Gaussian splatting framework while our method is based on the neural radiance field (NeRF) which can be naturally equipped with SDF representation for better surface reconstruction.
>
> **Q4: Does using SMPL pose parameters as LBS weights result in any artifacts?**
>
> Yes, when removing the non-rigid motion, using SMPL pose parameters alone as LBS weights can notably distort the body contours and desired patterns. But the appropriate non-rigid motion component can greatly alleviate this issue as in Figure 8.
>
> **Q5: What is the rendering speed of the proposed model?**
>
> As discussed in Section E, the MLPs used in our framework limit real-time applications, which is our most significant constraint. Currently, our method renders a 512x512 image with about ten seconds on a RTX 3090 GPU. However, please note that there is always a speed-accuracy tradeoff and our emphasis is on detail reproduction and model generalization. How to advance our inference speed with grid-based methods (e.g. triplane or hash grids) or 3D Gaussian Splatting framework is our future work.
>
> **Q6: Can you show the qualitative and quantitative results if the non-rigid Δx component is omitted?**
>
> As discussed in Q2, we denote the ablated model without Δx as "w / o locality" and present qualitative and quantitative results in Figure 8 and Table 3 (b) individually. Please see Q2 here and the ablation study part (Section 4.4) of our manuscript for more details.
>
> **Q7: Does considering non-rigid deformations help in accurately representing effects like the flapping of clothes?**
>
> Yes. As mentioned in Q2, considering non-rigid deformations can facilitate the pose-dependent effects which in turn improve the complicated clothing motion like the flapping of clothes.
>
> **We are looking forward to hearing more feedback from the reviewer.**

---

### Official Review · Reviewer_p8GG · 2024-11-04

**Soundness:** 2
**Presentation:** 3
**Contribution:** 2
**Rating:** 6
**Confidence:** 5

**Summary:**

This work presents a neural radiance field (NeRF) based network to learn human motions from monocular videos. The key argument is its method that can improve the local details of humans. They evaluated on ZJU-MoCap, ActorsHQ, SynWild, and various outdoor videos.

**Strengths:**

1. Human modeling from monocular video is an important task.
2. The method is clearly presented.
3. The code will be released.

**Weaknesses:**

1. Visual Quality. (1) In the demo video,  the gap between this work and SOTAs can not be clearly seen, especially for HumanNeRF. Samples at least on HumanRF/Synwild are also expected in the demo to better evaluate the universality of the work.  In the paper, the visual compassion is done on cases with simple textures and topologies. Other publicly available in-door datasets (e.g., AIST++, DNA-Rendering, MVHumanNet) with rich texture, and diverse/complex clothes should be evaluated. Also, the reconstruction difference with Vid2Avatar is hard to differentiate, which is also demonstrated in Table G, page 19.  (2)Whats the setting of the demo video is also not clear. Are they nvs, np, or just seen view reconstruction? The setting should be clarified.

2. Comparison to SOTAs. (1) Many state-of-the-art methods are missing in the comparison or discussion, such as MonoHuman in Tab.3 and the demo video, gaussian splatting based methods GaussianAvatar, 3DGS-Avatar (CVPR2024). (2) For the animation ability, it would be better to show driven poses that are out-of-distribution, like the MDM setting in MonoHuman.

3. Motivation of the method. The pipeline looks like a combination of different modules.  As the key of this work, details are improved by GNN. However, the motivation is not clear. What are the challenges of other feature extraction paradigms to model in detail? Why does GNN work? Why are other methods not working? If you add GNN to their implementation, will they also work?

**Questions:**

See Weakness.

---

> ### Author Response · Authors · 2024-11-22
> **Response to Reviewer p8GG --- Part 1**
>
> We thank the reviewer for reviewing our paper and providing valuable comments. We address the concerns in the common responses and below, and are open to further discussion and questions.
>
> **Q1: Video comparisons with SOTAs**
>
> We agree that the recent state-of-the-art methods make significant progress in neural avatar modeling and HumanNeRF provides a strong baseline for comparison. However, there are two key advantages of our method over HumanNeRF: 1) The locality encoding helps our method generalize better to challenging poses with improved details; 2) Our method can yield more plausible shape reconstruction.
>
> Given our conceptual advantages, it could be tricky to recognize our visual improvements in a playing video, especially in ZJU-Mocap sequences where the foreground is relatively dark. To this end, we pick six representative frames for more explicit visual comparisons in Figure J. Being consistent with the observations in L406, our method can yield better body shapes with less pattern distortion.
> Similar to Figure 7 of GaussianAvatar paper, we collect the challenging poses from AIST++ dataset as out-of-distribution animation inputs to further widen our visual improvements. Compared to HumanNeRF, our method can generate more consistent body parts while HumanNeRF distorts the left hand into discontinuous shape as shown in (2’09’’). Another example of out-of-distribution animation is illustrated in Figure 5.
>
> We also present novel pose animation results of loose cloth modeling, showcasing the complex interactions between the cloth and the human body. As shown in the “Comparison on ActorsHQ” and “Comparison on MVHumanNet” sections of the supplementary video, our method can output more satisfactory images while state-of-the-art baselines fail to preserve continuous shape contours.
>
> To illustrate our temporal consistency in producing shape geometry, we compare the normal map sequence with Vid2Avatar under the SynWild dataset. It is clear that our method can reproduce much sharper structured patterns (e.g. facial area) and delicate outlines (e.g. the pant boundary).
>
> Lastly, we want to emphasize that visually evaluating each method could be subjective due to different approaches having different failure modes and their weighting might differ from person to person. For example, Reviewer nejY acknowledges that our model demonstrates superior rendering performance compared to other approaches, particularly evident through enhanced normal rendering results. Thus to reach a more objective judgement, we perform comprehensive quantitative comparisons across totally 24 sequences of 6 datasets with 5 metrics and report detailed numbers in the appendix. Plus our advantages in geometry reconstruction, our method can clearly demonstrate its empirical effectiveness qualitatively and quantitatively.
>
> **Q2: Samples on more datasets are expected in video**
>
> Thank you for your suggestions. Please see the “Comparison on ActorsHQ”, “Comparisons on MVHumanNet” and “Comparison on SynWild” sections of the attached video for details. Due to time and computational constraints, we limit our comparison to Vid2Avatar on the SynWild dataset which is supposed to be the best baseline in geometry reconstruction. Results turn out that our method synthesizes much more geometric details than Vid2Avatar. We also improve time consistency over baselines under the settings of MVHumanNet and ActorsHQ datasets. Specifically, inputting challenging out-of-the-distribution poses in ActorsHQ sequences, baselines like HumanNeRF break the human body into many small pieces while our method preserves much more realistic shape contours. Our advantage remains valid in MVHumanNet sequences, showing our universality.

---

> ### Author Response · Authors · 2024-11-22
> **Response to Reviewer p8GG --- Part 2**
>
> **Q3: Reconstruction difference with Vid2Avatar is hard to differentiate**
>
> We discuss in Figure I and in L1068, that imperfect pseudo-ground truth hinders our ability to achieve fully accurate geometric evaluations quantitatively for ZJU-Mocap dataset as it smooths out the surface details and introduces unwanted artifacts (e.g. the ground surfaces). Thus the imperfect pseudo ground truth shapes benefit Vid2Avatar in most sequences with slightly higher Normal Consistency metrics. How to more accurately evaluate quantitative geometric results is still an open problem.
>
> Despite the marginal improvements due to inaccurate ground truth shapes, our method presents more faithful geometric details than Vid2Avatar visually. Specifically, our method can generate more clear facial structures in Figure 7 and Figure A, and can reconstruct more cloth wrinkles in the first row of Figure F. Our advantage in synthesizing fine-grained details is further enlarged when evaluating on the ActorsHQ dataset where clothes are more textured and actors are partially invisible. As shown in Figure N, our method still can reproduce vivid facial patterns and pant wrinkles under this challenging case. In comparison, Vid2Avatar simply smoothes out all these signals . We additionally present video results of the estimated normal maps in the 'Comparison on SynWild' section of the supplementary video, providing a comprehensive qualitative assessment. All these results demonstrate that our method significantly enhances Vid2Avatar by incorporating more geometric details.
>
> **Q4: Methods are missing in the comparison or discussion**
>
> Thank you for raising this question. Following your suggestions, we have included the results of MonoHuman into the table in “Common Response to Reviewers” for comparisons. Like the findings elsewhere, our method clearly surpasses MonoHuman by a large margin. Results of the newly added two baselines, GauHuman and PoseVocab, are also reported; please see the “Common Response to Reviewers” and the uploaded manuscript for more information. With results on presented datasets, our method outperforms the representative methods of canonical space rendering, locality encoding, and 3DGS driven modeling.
>
> **Q5: It would be better to show driven poses that are out-of-distribution**
>
> Thank you for your suggestion. Actually, as stated in L423, we have already shown the visual comparisons for out-of-distribution pose animation in Figure 5 and the supplementary video. Moreover, the animation results for the ActorsHQ sequence can also be regarded as out-of-distribution pose animation as the testing poses significantly differ from training poses. We have reorganized the supplementary video to make the demonstration more clear.
>
> **Q6: Motivation of the method**
>
> Inspired by previous methods, we leverage GNN as a simple way to process human skeletons and extract local features while we agree that there are other feature extraction paradigms as alternatives. But compared to other options to extract local features, such as time-dependent voxel planes [Wu et al. 2024], the GNN based feature extraction paradigm has the following benefits:
> (1) GNN propagates features within neighboring regions and can adaptively capture local patterns in multiple scales.
> (2) As inputs to GNN, the human skeleton is sparse and is able to yield decent expressiveness with limited computation complexity.
> (3) Pose estimation models can also work for testing poses and enable out-of-distribution pose animation.
>
> In comparison, time-dependent voxel planes struggle to model complex local relationships and cannot extend to novel pose animation due to fixed space-time resolutions. They are also dense in memory storage, which limits their capability in processing fine-grained details over time.
>
> Empirically, the effectiveness of GNN can be revealed through an ablation study. Specifically, we feed each skeleton node into its own MLP without feature propagation to extract features for each part, denoted as the “onlyPose” baseline. As listed in the table below, our full model which explicitly propagates nearby pose features constantly improves the “onlyPose” baseline.
>
> |            | PSNR  | SSIM   | LPIPS  |
> |:----------:|:-----:|:------:|:------:|
> | Novel View |       |        |        |
> | onlyPose   | 33.04 | 0.977  | 27.39  |
> | Full Model | **33.36** | **0.978** | **25.58**  |
> | Novel Pose |       |        |        |
> | onlyPose   | 33.20 | 0.976  | 27.57  |
> | Full Model | **33.44** | **0.977** | **26.34**  |
>
> However, note that our core technical contribution lies in integrating locality encoding into canonical space rendering and making solid conclusions through extensive experiments, rather than using a GNN for local feature extraction. As such, our conceptual approach can be adapted to other implementation frameworks, enabling potential extensions.
>
> [Wu et al. 2024] 4D Gaussian Splatting for Real-Time Dynamic Scene Rendering, CVPR 2024

---

> ### Comment · Reviewer_p8GG · 2024-11-27
>
> Thx authors for providing additional results and details. Based on this, I have raised the score accordingly.
>
> However, the remaining issues are:
>
> (1) While the later experiments presented during the rebuttal strengthen the effectiveness of the proposed framework, the technical contribution remains fundamentally incremental, appearing more as an engineering improvement.  I am not opposed to engineering-focused work, but to enhance the impact, the authors could explore the universality of the framework components further in the future, like whether the contribution components still work upon other base scene representations, what if the training number decreases, etc.,
>
> (2) A potential concern is that all samples used seem to have pure color or relatively simple textures, without coverage of more complex textures. Given the authors' claim of outperforming the sotas in nearly every aspect, testing with real-world, complex textures would better substantiate the claims. Additionally, the reconstruction quality in these scenarios could be further evaluated.
>
> (3) Be precise with claims. For example, the sample in Fig. 5 is not strictly out-of-distribution, as the target video includes a black lift-foot motion, which is not significantly dissimilar to the source.

---

> > ### Author Response · Authors · 2024-11-28
> > **Thank you for your response**
> >
> > Dear Reviewer p8GG,
> >
> > Thank you very much for your response and raising your score. We really appreciate it!
> >
> > For the technical contribution of our method, we would like to emphasize that our method contributes in two folds:
> > 1) Conceptually, we unify the canonical space rendering with locality encoding to improve generalization and adaptive detail synthesis in monocular settings while existing baselines prefer one to another with inferior results.
> > 2) Empirically, as mentioned in the previous responses, our method achieves the state-of-the-art results across novel view synthesis, novel pose animation and geometry reconstruction simultaneously.
> > However, we agree with the reviewer that exploring the universality of our method in other scene representations is valuable and could be our future work.
> >
> > Additionally, we follow the experimental setting of closest baselines (e.g. MonoHuman) and try our best to make a scientific conclusion with comprehensive comparisons shown in the "Justifying Experimental Protocols" part. Evaluating with more textured clothes and large scale motions is our future work.
> >
> > Thank you for your suggestions!

---

### Author Response · Authors · 2024-11-22
**Common Responses to Reviewers**

We thank all reviewers for the detailed feedback and the time committed. The reviewers recognize our studied task in this submission as important (Reviewer p8GG) and a long-standing problem in the community with different applications (Reviewer cJiu). Additionally, the method is clearly presented (Reviewer p8GG), novel (Reviewer nejY) and technically sound (Reviewer cJiu) with superior rendering performance (Reviewer nejY), showcasing its robustness and versatility (Reviewer nejY).

In this submission, our primary technical contribution is the integration of locality encoding into canonical space rendering, enabling simultaneous improvements in generality and adaptive detail, which is an aspect significantly overlooked by previous methods. **Empirically, we emphasize that one of our key advantages is to outperform state-of-the-art methods in novel view synthesis, novel pose animation, and shape reconstruction simultaneously for monocular human modeling. By comparing with seven SoTA methods in our original paper and two additional baselines, we make thorough evaluations on totally twenty four sequences across six datasets to ensure statistical significance.** We further justify our experimental setting in the following **Justifying Experimental Protocals** part in details.

As demonstrated in the tables within our appendix, different baselines exhibit unique advantages across various scenarios. Specifically, as shown in Figure 3, Figure 4 and Figure F, Vid2Avatar produces most reasonable 3D shapes among chosen baselines but often over-smoothes necessary fine-grained details. In parallel, other methods (e.g. HumanNeRF and 3DGS-Avatar) provide better rendering images than Vid2Avatar but yield very noisy surfaces. With numerous results reported here and in our manuscript, we conclude that our method consistently improves former algorithms in both novel image rendering and 3D shape reconstruction. Specifically, our method surpasses all the baselines on LPIPS with **over 10%** relative improvement across all scenarios and presents more faithfully geometric patterns. Our ablation study results further validate that including locality encoding and canonical space rendering in one single framework can facilitate the reproduction of both realistic shape contours and vivid fine-grained details. Questions asked by multiple reviewers are addressed below.

We hope that the provided clarifications and results, which demonstrate that the claims made in the paper hold over additional baselines and new sequences, can alleviate the remaining concerns. To ensure reproducibility, we will make the code of our method available upon acceptance.


**Clarifications on the supplementary videos (Reviewer p8GG, Reviewer cJiu)**

Thank you for pointing this out. To demonstrate generalization to unseen poses, all results are obtained under the novel pose setting, except for the MVHumanNet sequences. We use poses extracted from the AIST++ dataset to animate our avatar representation learned on Youtube sequences. For the results trained on remaining datasets, driven poses are provided by the official datasets including ZJU-Mocap, ActorsHQ and MVHumanNet. We further detail the dataset settings in Section B and add descriptions for the demo video in the **Summary of Supplementary Video** part of Section D in the appendix.

---

> ### Author Response · Authors · 2024-11-22
> **More Comparison Results**
>
> **Results on more challenging clothes with more baselines (Reviewer p8GG, Reviewer nejY, Reviewer cJiu)**
>
> Thank you for the constructive comments. As suggested by reviewers, we evaluate our method over three additional sequences (102107, 200173 and 204129) of the latest MVHumanNet (CVPR 2024) dataset to consider the varying degrees of clothing looseness. Additionally, as suggested, we include another 3DGS-based baseline, GauHuman (CVPR 2024), and a locality encoding method, PoseVocab (SIGGRAPH 2023), to further validate the importance of non-rigid deformation learning and the integration of canonical space with locality encoding. For MVHumanNet sequences, since the subjects only turn around near the end of each sequence, we use all frames for training to capture the whole body and thus omit the novel pose evaluation. For each baseline, we achieve results through their public source code with hyper-parameters as specified in their papers.
>
> We illustrate the comparison examples in Figure F, Figure G and Figure H of the uploaded revision. As reported in the table below and detailed in Table G, our method achieves better quantitative results than all baselines in terms of novel view synthesis.
>
> |             | PSNR  | SSIM  | LPIPS |
> |:-----------:|:-----:|:-----:|:-----:|
> | HumanNeRF   | 23.07 | 0.956 | 55.45 |
> | MonoHuman   | 22.64 | 0.956 | 62.21 |
> | PoseVocab   | 20.85 | 0.949 | 69.42 |
> | Vid2Avatar  | 22.98 | 0.958 | 60.32 |
> | 3DGS-Avatar | 23.74 | 0.958 | 57.90 |
> | GauHuman    | **23.76** | 0.957 | 72.90 |
> | Ours        | 23.56 | **0.960** | **47.18** |
>
> Specifically, our method achieves over a 17.5% relative improvement in LPIPS compared to the second-best baseline (HumanNeRF). Together with the results assembled in the ActorsHQ dataset, we have utilized seven sequences to convincingly demonstrate our robustness to loose-fitting clothes. Please note that our method improves HumanNeRF with a much smaller network size: 1.2M (ours) vs 60M (HumanNeRF). Similarly, we achieve better results than MonoHuman whose number of trainable parameters is over 74M.
>
> Additionally, note that, we outperform the monocular video focused methods (e.g. HumanNeRF and 3DGS-Avatar) that in turn outperform the remaining multiple view based baselines (e.g. NPC, PM-Avatar) as shown in Table 2 and Table 3 (a). Additionally, as summarized in Table 2 of the 3DGS-Avatar paper, this baseline achieves a better balance between quality and speed compared to other 3DGS-driven algorithms. It also incorporates pose-dependent cloth deformation, making it conceptually the most effective among 3DGS-based methods.
>
> Thus we categorize our baselines into three classes: 1) NeRF based methods with only canonical space rendering, including HumanNeRF, MonoHuman and Vid2Avatar; 2) NeRF based methods with only locality encoding, including PoseVocab, NPC and PM-Avatar; 3) 3DGS driven methods, including 3DGS-Avatar, GoMAvatar and GauHuman. Given video inputs, these three categories of methods can best represent the existing neural avatar modeling pipelines.
>
> [Qian et al., 2024] 3DGS-Avatar: Animatable Avatars via Deformable 3D Gaussian Splatting, CVPR 2024.

---

> ### Author Response · Authors · 2024-11-22
> **Justifying Experimental Protocols --- Part 1**
>
> **Comparison of Experimental Settings for NeRF based baselines (Reviewer p8GG, Reviewer nejY, Reviewer cJiu)**
>
> As mentioned above, we have incorporated sequences from the MVHumanNet dataset into our testbed. To further validate the comprehensiveness of our evaluation protocol, we provide a detailed comparison of the experimental settings for the NeRF based baselines below.
>
> |                         |                                                    Chosen datasets for image synthesis                                                   | Sequence Numbers |                                                                                                         Chosen baselines                                                                                                        | Baseline Number |
> |:-----------------------:|:----------------------------------------------------------------------------------------------------------------------------------------:|:----------------:|:-------------------------------------------------------------------------------------------------------------------------------------------------------------------------------------------------------------------------------:|:---------------:|
> |  HumanNeRF (CVPR 2022)  |                                         ZJU-MoCap (CVPR 2021), self-captured data, YouTube video                                         |        11        |                                                                                     Neural Body (CVPR 2021), HyperNeRF (SIGGRAPH Asia 2021)                                                                                     |        2        |
> |  MonoHuman (CVPR 2023)  |                                                   ZJU-MoCap (CVPR 2021), YouTube video                                                   |         8        |                                                                                Neural Body (CVPR 2021), HumanNeRF (CVPR 2022), NeuMan (ECCV 2022)                                                                               |        3        |
> |  Vid2Avatar (CVPR 2023) |                        MonoPerfCap (SIGGRAPH 2018), 3DPW (ECCV 2018), NeuMan (ECCV 2022), SynWild (self-captured)                        |        15        |                                        PointRend (CVPR 2020), RVM (WACV 2022), Ye et al. (CVPR 2022), ICON (CVPR 2022), SelfRecon (CVPR 2022), HumanNeRF (CVPR 2022), NeuMan (ECCV 2022)                                        |        7        |
> |     NPC (ICCV 2023)     |                                 Human3.6M (ICCV 2011), MonoPerfCap (SIGGRAPH 2018), ZJU-Mocap (CVPR 2021)                                |        10        |                                                   NeuralBody (CVPR 2021), Anim-NeRF (ICCV 2021), A-NeRF (NeurIPS 2021), TAVA (ECCV 2022), DANBO (ECCV 2022), ARAH (ECCV 2022)                                                   |        6        |
> |  PM-Avatar (ICLR 2024)  |                                 Human3.6M (ICCV 2011), MonoPerfCap (SIGGRAPH 2018), ZJU-MoCap (CVPR 2021)                                |        13        |                NeuralBody (CVPR 2021), Anim-NeRF (ICCV 2021), A-NeRF (NeurIPS 2021), TAVA (ECCV 2022), DANBO (ECCV 2022), ARAH (ECCV 2022), HumanNeRF (CVPR 2022), MonoHuman (CVPR 2023), Vid2Avatar (CVPR 2023)                |        9        |
> |           Ours          | MonoPerfCap (SIGGRAPH 2018), ZJU-Mocap (CVPR 2021), SynWild (CVPR 2023), ActorsHQ (SIGGRAPH 2023), MVHumanNet (CVPR 2024), Youtube video |        24        | HumanNeRF (CVPR 2022), MonoHuman (CVPR 2023), Vid2Avatar (CVPR 2023), PoseVocab (SIGGRAPH 2023), NPC (ICCV 2023), PM-Avatar (ICLR 2024), 3DGS-Avatar (CVPR 2024), GoMAvatar (CVPR 2024), GauHuman (CVPR 2024) |        9       |
>
> We further paste the comparisons of evaluation protocols for 3DGS-based methods in the following section.

---

> ### Author Response · Authors · 2024-11-22
> **Justifying Experimental Protocols --- Part 2**
>
> **Comparison of Experimental Settings for 3DGS based baselines (Reviewer p8GG, Reviewer nejY, Reviewer cJiu)**
>
> |                         |                                                    Chosen datasets for image synthesis                                                   | Sequence Numbers |                                                                                                         Chosen baselines                                                                                                        | Baseline Number |
> |:-----------------------:|:----------------------------------------------------------------------------------------------------------------------------------------:|:----------------:|:-------------------------------------------------------------------------------------------------------------------------------------------------------------------------------------------------------------------------------:|:---------------:|
> | 3DGS-Avatar (CVPR 2024) |                                             PeopleSnapshot (CVPR 2018), ZJU-MoCap (CVPR 2021)                                            |        10        |                                                         NeuralBody (CVPR 2021), HumanNeRF (CVPR 2022), ARAH (ECCV 2022), MonoHuman (CVPR 2023), Instant-NVR (CVPR 2023)                                                         |        5        |
> |  GoMAvatar (CVPR 2024)  |                                     PeopleSnapshot (CVPR 2018), ZJU-MoCap (CVPR 2021), YouTube video                                     |        12        |                                           Anim-NeRF (arXiv 2021), NeuralBody (CVPR 2021), HumanNeRF (CVPR 2022), NeuMan (ECCV 2022), MonoHuman (CVPR 2023), InstantAvatar (CVPR 2023)                                           |        6        |
> |   GauHuman (CVPR 2024)  |                            MonoCap (CVPR 2020+ACM ToG 2021), ZJU_MoCap (CVPR 2021),  DNA-Rendering (ICCV 2023)                           |        12        |          PixelNeRF (CVPR 2021), Neural Body (CVPR 2021), Anim-NeRF (ICCV 2021), NHP (NeurIPS 2021), AniSDF (arXiv 2022), HumanNeRF (CVPR 2022), DVA (SIGGRAPH 2022), InstantNVR (CVPR 2023), InstantAvatar (CVPR 2023)          |        9        |
> |           Ours          | MonoPerfCap (SIGGRAPH 2018), ZJU-Mocap (CVPR 2021), SynWild (CVPR 2023), ActorsHQ (SIGGRAPH 2023), MVHumanNet (CVPR 2024), Youtube video |        24        | HumanNeRF (CVPR 2022), MonoHuman (CVPR 2023), Vid2Avatar (CVPR 2023), PoseVocab (SIGGRAPH 2023), NPC (ICCV 2023), PM-Avatar (ICLR 2024), 3DGS-Avatar (CVPR 2024), GoMAvatar (CVPR 2024), GauHuman (CVPR 2024) |        9       |
>
> The tables detailing experimental settings clearly demonstrate that our method conducts the most comprehensive evaluations compared to all the chosen baselines. Specifically, we compare against the largest number of baselines across the most extensive and recently released datasets and sequences. Furthermore, apart from GauHuman, ours is the only method that leverages publicly available datasets released after 2023 for evaluation. Combined with the significant empirical improvements achieved on each dataset, we can confidently conclude that our method improves state-of-the-art approaches in novel image rendering and 3D shape reconstruction which highlights the effectiveness of our proposed method.

---

> ### Author Response · Authors · 2024-11-27
> **Adding results of MonoHuman**
>
> Dear reviewers,
>
> We have updated Table G in the appendix and the table here to include quantitative results for MonoHuman on sequences from MvHumanNet (CVPR 2024).
>
> Please note that this can further reveal our universal advantages over baselines across varying degrees of cloth looseness. Alongside the visual comparisons in Figures F, H, and I, and the results from ActorHQ (SIGGRAPH 2023) presented in Tables E, F, and Figures N, R, the comprehensive evaluation demonstrates that our method consistently outperforms all three classes of state-of-the-art methods:
>
> + **NeRF-based methods with canonical space rendering**: HumanNeRF, MonoHuman, and Vid2Avatar.
> + **NeRF-based methods with locality encoding**: PoseVocab and PM-Avatar.
> + **3DGS-based methods**: 3DGS-Avatar and GauHuman.
>
> These results underscore the superior performance of our method across diverse scenarios.
>
> We deeply value reviewers' comments and strongly encourage reviewers to share any additional feedback or insights regarding these updates. We eagerly look forward to the reviewers' detailed review and constructive discussions.

---

> > ### Author Response · Authors · 2024-11-29
> > **Looking forward to further response**
> >
> > Dear Reviewer nejY and Reviewer cJiu,
> >
> > Thank you for taking the time to provide such valuable feedback. As the weekend approaches, we wanted to kindly check if you have any additional concerns or points for discussion. We are looking forward to any further feedback you may have on our revised manuscript.
> >
> > Thank you once again!

---

### Author Response · Authors · 2024-11-22
**Revision Uploaded**

We thank all of the reviewers and ACs for their time and effort in providing these helpful suggestions. Based on these reviews, we have revised our paper and uploaded the new version. Changes are highlighted in blue. To summarize, we have made the following changes to our paper and supplementary material:

1. Revising Table 1 and the “Modeling Articulated Avatars” part of Section 2 to summarize the difference between our method and newly added baselines, including GauHuman and GaussianAvatar.
2. Updating the manuscript to clarify the experiment comparisons on the ActorsHQ dataset (L418-L422) and discussions on out-of-distribution motions (L423-L426).
3. Adding two paragraphs in Section C to detail the advantages of using GNN to encode locality information.
4. Making a new paragraph (**Summary of Supplementary Video**) in Section D to summarize the contents of the supplementary video for better demonstration.
5. Adding one figure in Figure J to make visual comparisons across six consecutive frames from the demo video on the ZJU-Mocap dataset.
6. Adding one figure in Figure M to illustrate the non-rigid effects and validate the importance of locality encoding.
7. Adding one figure in Figure N to compare the shape meshes reconstructed from ActorsHQ dataset.
8. Adding results in Table G for quantitative comparisons and results in Figure F, Figure G, Figure H for qualitative comparisons on the MVHumanNet dataset.
9. Updating Tab. 3 (a) and adding Tab. E and Tab. F to incorporate one more sequence of ActorsHQ dataset and include more comparison baselines. The corresponding figure is shown in Figure R.
10. Adding chapters in the supplementary video to compare under the dataset setting of SynWild, ActorsHQ, MVHumanNet.

---

### Author Response · Authors · 2024-11-25
**Thanks for the reviews**

Dear reviewers,
﻿

As the discussion period is close to end, we thank you for your time and efforts on reviewing our paper with numerous constructive suggestions, from detailed expositions to comparisons with more recent methods on the latest dataset. We really appreciate them! During the author-reviewer discussion period, the authors have tried to address reviewers’ every comment with great efforts. In particular, we significantly consolidated our experiment part by adding comparison results suggested by all reviewers. With the additional experiments, we can confidently conclude the empirical superiority of our method over state-of-the-art baselines. Lastly, the authors commit to releasing the code upon acceptance.
﻿

Based on our updated manuscript, could the reviewers re-evaluate our submission? We are also open to new discussions.

---

### Author Response · Authors · 2024-11-26
**Updating the "Comparison on SynWild" part of video**

Dear reviewers,

To further demonstrate our advancements in producing shape geometry, we add a comparison with GoMAvatar, the 3DGS-based state-of-the-art method in shape reconstruction, alongside the NeRF-based Vid2Avatar method. Please refer to the **"Comparison on SynWild"** section of the supplementary video, where we compare the normal map sequences of our method with both approaches on the SynWild dataset. The results clearly show that our method outperforms both NeRF-based and 3DGS-based state-of-the-art methods, delivering sharper structured patterns and finer details.

We would greatly appreciate any additional feedback or insights you may have regarding the updates. We look forward to your review and continued discussions.

---

### Author Response · Authors · 2024-12-04
**Thank you for your thorough reviews**

Dear Reviewers and ACs,

First, thank you again for your time and efforts to conduct such thorough and helpful discussions. To echo your professional comments, we  were diligently replying reviewers's questions and have significantly improved our submission in aspect of providing more comprehensive evaluation results and offering further clarifications  as summarized before. With these new updates, we are more confident about our scientific conclusion that the unification of the canonical space rendering and local deformation can significantly improve baselines in novel view synthesis, novel pose animation and shape reconstruction simultaneously. While this technical upgrade is simple and straightforward, we would emphasize that it represents a significant oversight by the research community and we make the first step to unify them together which is a clear conceptual technical contribution. Also our complete experiments with notable improvements further demonstrate our empirical advantages widely.

To reveal our effectiveness for the clothing modeling with real-world, complex textures, we provide quantitative comparisons here and will attach visual presentation in the revision. Following the setting of MonoHuman and PM-Avatar which are best baselines equipped with canonical space rendering and locality encoding respectively, we pick one sequence ("Weipeng") from MonoPerfCap dataset and two online Youtube sequences ("Way2sexy" and "Invisible") for comparisons. The exemplified figures are listed below:
1. Weipeng: The second row of Figure 4 in PM-Avatar (https://openreview.net/pdf?id=5t44vPlv9x).
2. Way2sexy: https://www.youtube.com/watch?v=gEpJDE8ZbhU.
3. Invisible: https://www.youtube.com/watch?v=ANwEiICt7BM.

Below, the comparison results on these three subjects wearing clothes with intricate textures or structured patterns furthermore highlight our robustness to complex garments.

|           | LPIPS |  FID  |  KID  |
|:---------:|:-----:|:-----:|:-----:|
| MonoHuman | 37.05 | 65.14 | 24.29 |
| PM-Avatar | 47.36 | 93.67 | 64.19 |
|    **Ours**   | **32.48** | **59.22** | **22.76** |


Additionally, please note that we generally follow the settings of latest baselines (MonoHuman, Vid2Avatar, 3DGS-Avatar and GoMAvatar) to perform fair comparisons including the definition of novel pose rendering. Except the video results on Youtube sequences, the video on ActorsHQ sequences provides another representative demo for the out-of-distribution animation because there is no squatting action in the training data. As discussed at Section E,  while no current methods can synthesize good results for loose clothing and extremely out-of-distribution poses, we can produce more visually appealing results than baselines.

Lastly, we sincerely thank Reviewer p8GG for giving detailed suggestions to enhance our experimental assessment. We also deeply appreciate Reviewer cJiu for the further suggestions on exposition and recognizing **the great potential of our method**. We would incorporate your comments into our revision and future research directions. It will also be great to hear more feedbacks from Reviewer nejY in the updated official comments afterwards.

---

### Meta-Review · Area_Chair_ATNZ · 2024-12-20

**Metareview:**

This paper introduces locality sensitive avatars which learns human motions from monocular videos. It decompose motion into canonical space and corresponding offset to non-rigid effects. A GNN is applied to model the non-rigid effects so as to keep fine-grained, locality-sensitive details. The proposed methods are tested on several datasets and achieved state-of-the-art results.

The proposed method studies an important problem as acknowledged by the reviewers. The proposed method is tested on several benchmarks and achieved state-of-the-art results. The original submission lacked experiments on some challenging dataset and comparisons with some existing works, but the authors addressed these concerns during the discussion. The paper is overall not so well written as pointed out by reviewer cJiu which makes it challenging to follow.

After discussion, all reviewers agree that this paper is above borderline, so I tend to accept this paper.

**Additional Comments On Reviewer Discussion:**

The authors did a great job in addressing concerns raised by reviewers. They follow the requirements from the reviewers to include experiments on new benchmarks, adding new comparisons with existing methods and clarify misleading statements. The authors addressed all major concerns from the reviewers and all reviewers raised their ratings for this paper.

---

### Decision · Program_Chairs · 2025-01-22

Accept (Poster)